# Biofilm formation in the lung contributes to virulence and drug tolerance of *Mycobacterium tuberculosis*

Poushali Chakraborty[1], Sapna Bajeli[1], Deepak Kaushal [2], Bishan Dass Radotra[3] & Ashwani Kumar [1,4✉]

Tuberculosis is a chronic disease that displays several features commonly associated with biofilm-associated infections: immune system evasion, antibiotic treatment failures, and recurrence of infection. However, although *Mycobacterium tuberculosis* (Mtb) can form cellulose-containing biofilms in vitro, it remains unclear whether biofilms are formed during infection in vivo. Here, we demonstrate the formation of Mtb biofilms in animal models of infection and in patients, and that biofilm formation can contribute to drug tolerance. First, we show that cellulose is also a structural component of the extracellular matrix of in vitro biofilms of fast and slow-growing nontuberculous mycobacteria. Then, we use cellulose as a biomarker to detect Mtb biofilms in the lungs of experimentally infected mice and non-human primates, as well as in lung tissue sections obtained from patients with tuberculosis. Mtb strains defective in biofilm formation are attenuated for survival in mice, suggesting that biofilms protect bacilli from the host immune system. Furthermore, the administration of nebulized cellulase enhances the antimycobacterial activity of isoniazid and rifampicin in infected mice, supporting a role for biofilms in phenotypic drug tolerance. Our findings thus indicate that Mtb biofilms are relevant to human tuberculosis.

[1] Council of Scientific and Industrial Research, Institute of Microbial Technology, Chandigarh, India. [2] Southwest National Primate Research Center, Texas Biomedical Research Institute, San Antonio, TX, USA. [3] Department of Histopathology, Postgraduate Institute of Medical Education and Research, Chandigarh, India. [4] Academy of Scientific and Innovative Research (AcSIR), Ghaziabad, Uttar Pradesh, India. ✉email: ashwanik@imtech.res.in

Bacterial biofilms are often associated with chronic infections in humans[1]. Biofilm formation inside the host bestows several benefits to the pathogens, such as protection from the host immune system[2], treatment failures even with higher doses of antibiotics[3], and recurrence of infection[4]. Importantly, primary mycobacterial infections, tuberculosis (TB), and leprosy display these characteristics. *Mycobacterium tuberculosis* (Mtb) causes TB. During infection, Mtb remains hidden from the immune system[5,6], and its treatment utilizes prolonged administration of multiple drugs[7]. Notably, the recurrence of TB is often seen in patients who have received anti-TB treatment[8]. These features resemble microbial biofilm infection; however, whether Mtb forms biofilms during infection remains unknown. Notably, Mtb has a natural tendency to adhere to surfaces and forms cords in the culture medium[9,10]. Incidentally, the cording behavior is associated with virulence and pathogenicity of Mtb[9]. Biofilms are well known to harbor drug-tolerant bacteria. In agreement with this behavior, Ojha et al. have demonstrated that Mtb pellicle biofilms contribute to the exhibition of in vitro drug tolerance[11]. Further, building on these observations, Ackart et al. have recently shown that Mtb cells form biofilms adhered to the surface of the culture dish, harboring drug-tolerant bacilli encased in an extracellular matrix derived from lysed human leukocytes[12]. Interestingly, the dispersion of Mtb communities using small molecules reinstates drug sensitivity in Mtb cells, suggesting that phenotypic drug tolerance in these models is primarily due to community behavior[13,14]. Subsequent studies suggested that keto mycolic acids play a critical role in mycobacterial biofilm formation[15]. It was also suggested that the free mycolic acids could be released through enzymatic hydrolysis of trehalose dimycolate[16]. Recently, we have demonstrated that thiol reductive stress induces the formation of surface adherent Mtb biofilms[17]. These biofilms also harbor drug-tolerant Mtb cells. Given the tendency to form aggregates as well as biofilms in cultures and the chronic nature of TB, it could be hypothesized that Mtb forms biofilms inside the host.

One of the significant hurdles in identifying the presence of Mtb biofilms in vivo is the absence of a suitable biomarker. We have recently demonstrated that cellulose is a vital structural component of Mtb biofilms[17]. Cellulose is a glucose polymer, wherein the $\beta(1 \rightarrow 4)$ linkage connects the glucose units. Given that presence of cellulose was a surprising finding, several approaches ranging from staining with Calcofluor White (CW) to purification and characterization using X-ray powder diffraction and FTIR analysis were employed to conclude that cellulose is a component of Mtb biofilms[17]. Cellulose is synthesized in bacterial cells using a multiprotein complex called cellulose synthase. The core catalytic activity of cellulose synthase resides in the BcsA and BcsB proteins[18,19]. Several mycobacterial species, such as *M. neoaurum*, *M. cosmeticum*, etc., contain genes encoding for components of the cellulose synthase. This raises the possibility that mycobacterial species could use cellulose as a structural component of biofilms. Although cellulose could be detected in the Mtb biofilms, the Mtb genome does not seem to encode either homolog of canonical BcsA or BcsB. However, the role of cellulose in the mycobacterial biofilms is supported by a recent study demonstrating that the overexpression of cellulase in *M. smegmatis*[20] prevents pellicle formation. The role of cellulose in Mtb biofilms is further supported by the observation that the Mtb genome encodes for a few cellulases[21,22]. These cellulases have been biochemically and structurally characterized, but their biological functions are not precisely defined. Since cellulose is absent in humans and other animal models, its presence around the Mtb cells in infected animals or human tissues could indicate the presence of Mtb biofilms in vivo[23,24].

In this study, we have utilized several biochemical and biophysical methods to characterize the nature of extracellular polymeric substances (EPS) of slow and fast-growing nontuberculous mycobacteria (NTM). Furthermore, we have tested the hypothesis that Mtb forms biofilms inside the host lung tissue. Specifically, we have utilized mice and non-human primate models of TB to answer this question. We have also analyzed whether Mtb forms biofilms in the human lungs. We have also investigated whether Mtb biofilms protect the resident bacilli from the host immune system and antimycobacterial agents.

## Results

**Thiol reductive stress induces biofilm formation in *M. avium*, *M. abscessus* and *M. fortuitum*.** NTM are mycobacteria that do not cause TB or leprosy. To study the general characteristics of mycobacterial biofilms, we selected representatives of fast-growing NTM, namely *M. abscessus* (Mab) and *M. fortuitum* (Mfo). At the same time, *M. avium* (Mav) was used as a representative of slow-growing NTM. Importantly, Mav and Mab are clinically relevant, as well. NTM are known to aggregate[23] at the air–media interface, forming the typical pellicle biofilms (Supplementary Fig. 1a–c). Earlier, we have shown that Mtb forms biofilms in response to thiol reductive stress[17]. Although NTM readily make pellicle biofilms, it is unknown whether thiol reductive stress induces biofilm formation in NTM similar to Mtb. Towards this, we exposed cultures of Mav, Mab, and Mfo to 6 mM Dithiothreitol (DTT), β-mercaptoethanol (βME) and oxidized DTT. Interestingly, DTT treatment resulted in the formation of surface adherent, submerged biofilms for all the three NTM tested in this study (Fig. 1a). Quantitation of the biofilm formation using crystal violet assay also confirmed that thiol reductive stress manifested by DTT induces biofilm formation in these NTM (Fig. 1b). However, cell impermeant thiol reductant βME and oxidized DTT were unable to induce biofilm formation in the three NTM (Fig. 1a, b). We also analyzed whether DTT causes thiol-reductive stress in Mav, Mab, and Mfo cultures similar to Mtb cultures. Towards this, Mav, Mab, and Mfo cultures were exposed with 6 mM DTT and intracellular thiol levels were measured. We observed that only DTT exposure results in intracellular thiol stress in Mav, Mab, and Mfo (Supplementary Fig. 2a–c), again pointing out that intracellular thiol reductive stress induces biofilm formation. We also tested whether DTT exposure induces biofilm formation in Mav, Mab, and Mfo cultures agitated at slow speed. We observed that robust biofilms were formed at the media air interface in the cultures of Mav, while thin biofilms formed with Mfo cultures under shaking conditions; however, in Mab cultures, an increase in the turbidity was observed, but biofilms did not form at the liquid-air interface (Fig. 1c). To estimate DTT's minimum concentration required to induce biofilm formation in the standing cultures, we independently exposed all the three NTM cultures to increasing DTT concentrations (0.125 mM to 8 mM). All the three NTM, namely, Mab, Mav, and Mfo, required at least 2 mM DTT to initiate biofilm formation (Supplementary Fig. 3a–c). Since biofilms are known to harbor drug-tolerant microorganisms, we analyzed if thiol reductive stress-induced biofilms harbor drug-tolerant bacteria. Towards this, NTM biofilms were treated with 1×, 10× and 100× MIC concentrations of Bedaquiline, and mycobacterial survival was assessed. Interestingly, we did not observe a change in bacterial survival even after treatment with 100× MIC of the drug, suggesting the presence of drug-tolerant bacteria in the biofilms (Fig. 1d).

**Mav, Mab and Mfo biofilms contain polysaccharide-rich extracellular matrix.** Biofilms are encased within a self-synthesized matrix of EPS. This EPS is considered to be the hallmark of biofilms. Earlier, studies have shown that mycolic acids constitute a

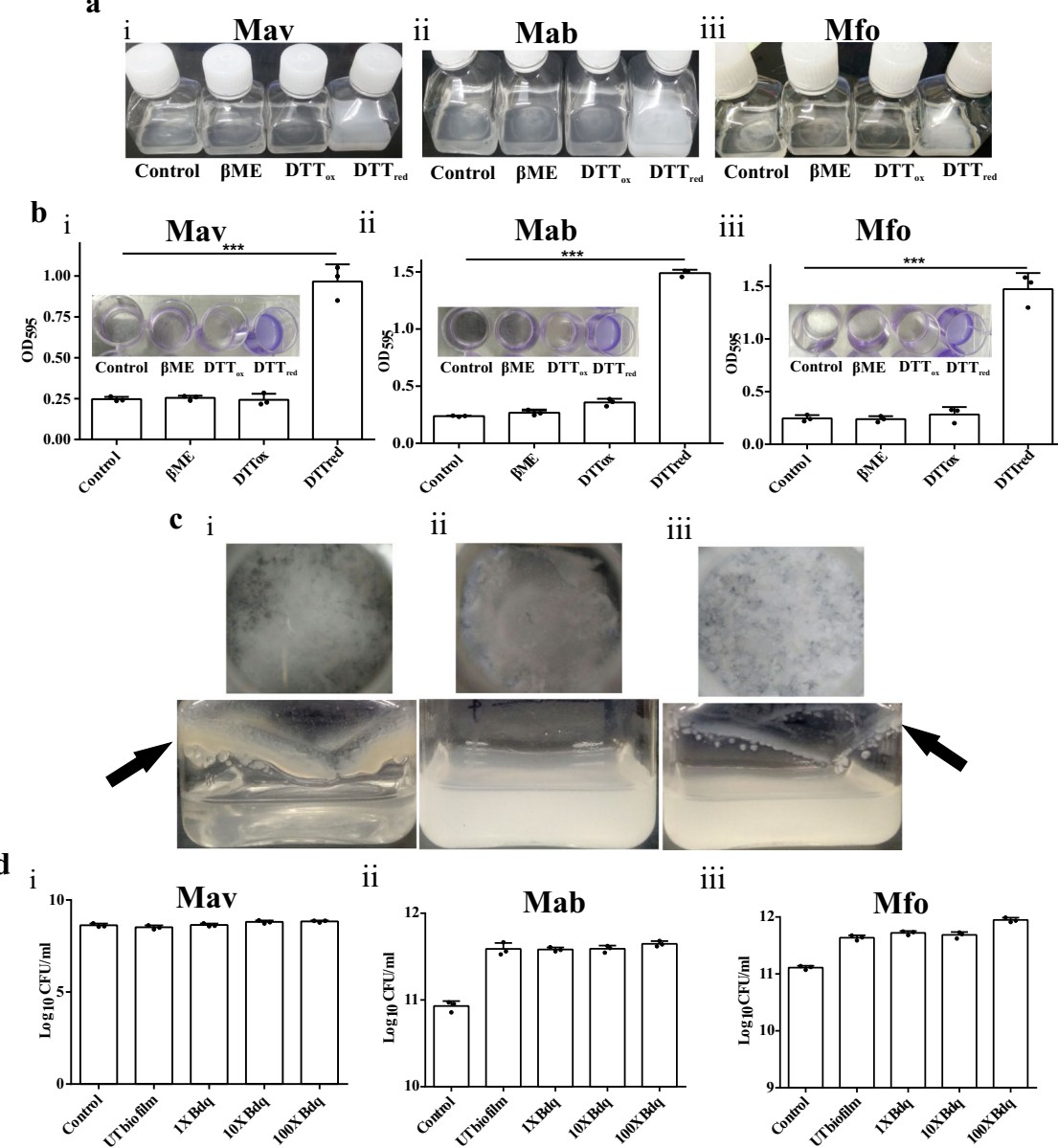

**Fig. 1 Thiol reductive stress leads to drug-tolerant biofilm formation in NTMs. a** Exponential cultures of Mav (**i**), Mab (**ii**) and Mfo (**iii**) were exposed to 6 mM of βME, oxidized, or reduced DTT individually. Only exposure to reduced DTT resulted in biofilm formation. **b** Cultures of Mav, Mab, and Mfo were exposed to 6 mM of DTT to induce biofilm formation. CV assays were performed to quantitate Mav (**i**), Mab (**ii**), and Mfo (**iii**) biofilms. **c** Thiol reductive stress-induced submerged biofilm of Mav (**i**), Mab (**ii**) and Mfo (**iii**) in standing culture (upper panel) and under slow shaking (lower panel). Biofilms formed are indicated with black arrows. **d** Colony forming units of Mav (**i**), Mab (**ii**) and Mfo (**iii**) were estimated after treatment with 1×, 10× and 100× MIC of Bedaquiline by degrading the biofilms of the NTMs using cellulase and plating them on 7H11 agar after serially diluting them. The column bar graphs were plotted using GraphPad Prism 6 and represented as mean (±s.e.m). Statistical significance was determined using Student's t-test (two tailed). For **b** (**i**), (**ii**) and (**iii**), ***$P = 0.0003$, ***$P < 0.0001$ and ***$P = 0.0002$ respectively. All data are representative of three independent biological experiments performed in triplicates. All source data are provided as a Source Data file.

significant component of the pellicle biofilms of Mtb[11], while we have observed that TRS induced Mtb biofilms are rich in polysaccharides[17]. However, the biochemical nature of EPS of the NTM biofilms remains poorly characterized. In an effort to understand the nature of EPS in NTM biofilms, we analyzed EPS of NTM colonies through plating on media containing Congo Red (CR) and Coomassie Brilliant Blue (CBB)[25]. CR could label large complexes like cellulose or amyloid structures, while CBB stains proteins in the colony biofilms. We observed that NTM colonies take up CR on the 7H11 plates, suggesting the presence of a polysaccharide-rich or amyloid-rich EPS (Supplementary Fig. 4a–c). We also induced

NTM biofilms using DTT in chamber slides to enable microscopic examination. Biofilms were stained with specific dyes and analyzed using confocal microscopy. Staining of Mav biofilms with propidium iodide revealed the presence of extracellular DNA in the matrices of these surface-attached NTM biofilms (Fig. 2a). The extracellular DNA might aid in anchoring bacilli to the substratum. SYPRO Ruby staining of the biofilms revealed the presence of matrix proteins in the EPS of Mav biofilms. Importantly, Mav biofilms stained profusely with Texas red hydrazide depicting the extensive presence of polysaccharides in spaces between the bacterial masses. These data suggest that large quantities of

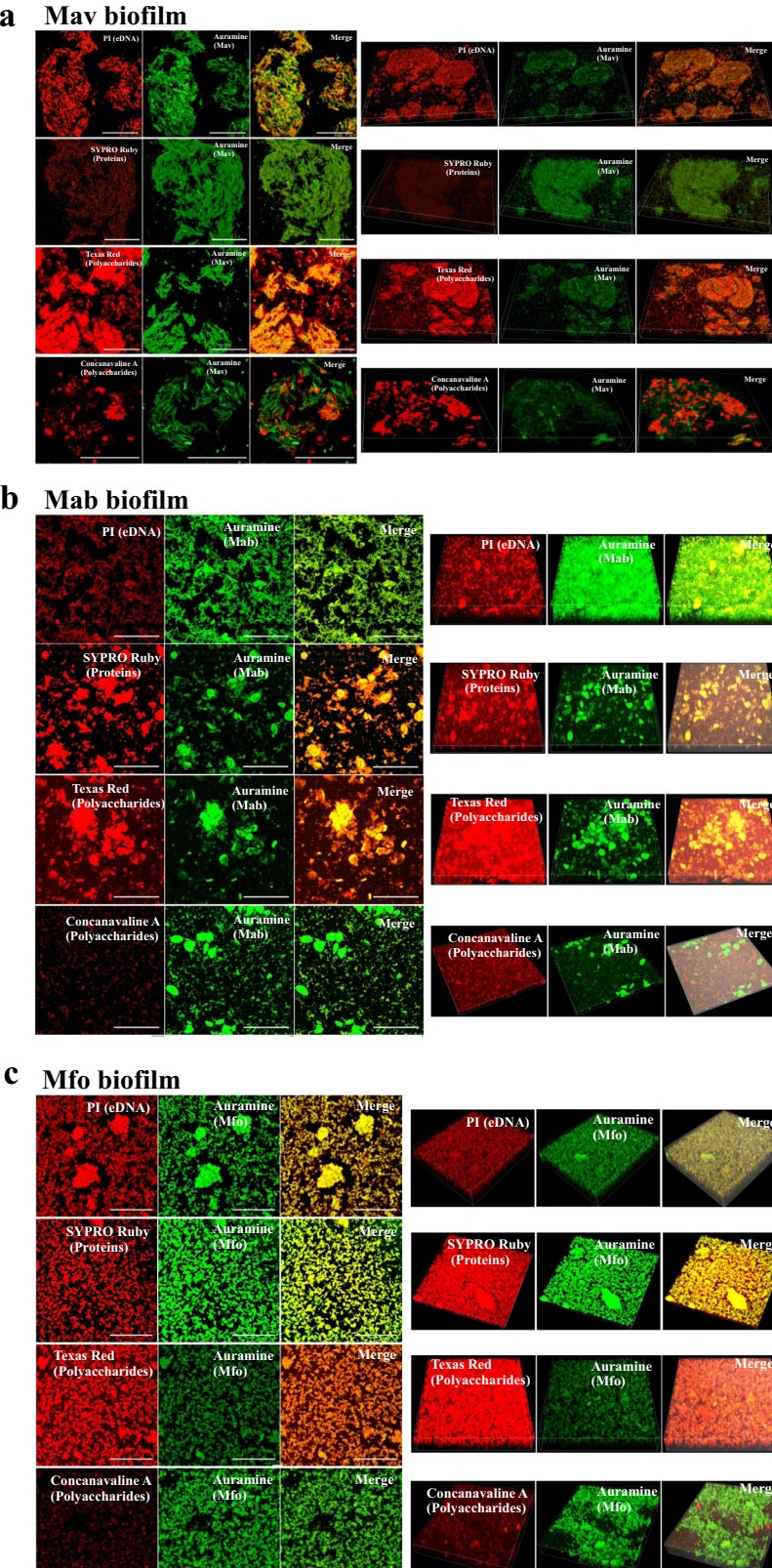

**Fig. 2 Characterization of thiol reductive stress-induced biofilm matrices in NTMs. a–c** To characterize the chemical nature of the extracellular matrix of thiol reductive stress-induced biofilms, Mav (**a**), Mab (**b**) and Mfo (**c**) cultures were subjected to 6 mM DTT for 29 h and then stained with PI (for eDNA), SYPRO Ruby (for proteins), Texas red (for polysaccharides) and Concanavalin A (for α-glucopyranosyl and α-mannopyranosyl residues). NTM cells were stained with phenolic Auramine O – Rhodamine B and analyzed using CLSM. All data are representative of three independent biological experiments. Scale bars indicate 10 µm.

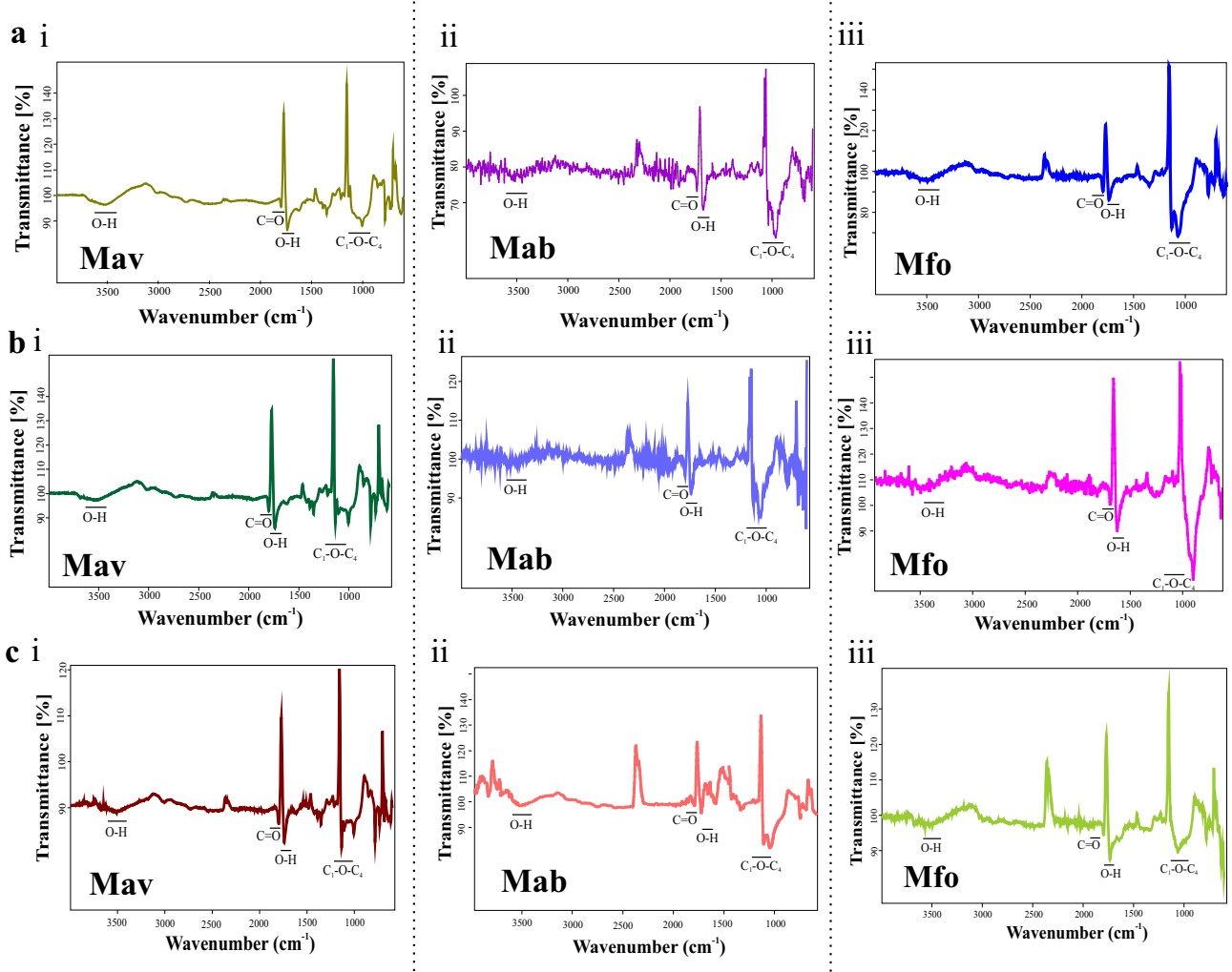

**Fig. 3 Characterization of biofilm matrix polysaccharides. a–c** Cellulose was purified from submerged biofilms (**a**), pellicle biofilms (**b**) and macrocolony biofilms (**c**) of Mav (**i**), Mab (**ii**) and Mfo (**iii**) and then subjected to FTIR analysis. All the experiments are representatives of at least three biological experiments performed with three technical replicates each.

polysaccharides are present between bacterial cells, and thus polysaccharides are the major components of the extracellular matrix. The polysaccharide-rich nature of the three NTM is similar to Mtb biofilms. The biofilms were also strained with Concanavalin A that bound explicitly to α-D glucopyranosyl and mannopyranosyl residues (Fig. 2a). A similar staining pattern was observed with the biofilms of Mab and Mfo (Fig. 2b, c).

**Cellulose is a structural component of Mav, Mab and Mfo biofilms**. To further investigate the chemical composition of the biofilm matrix, we washed the biofilms of Mav, Mab and Mfo with PBS and then treated them with different EPS degrading enzymes like Cellulase, Proteinase K, α-Amylase, Lipase and DNaseI. We observed that Cellulase and Proteinase K disintegrated the biofilms of NTM. However, DNaseI had minor effects, whereas Lipase and α-Amylase did not affect Mav, Mab and Mfo biofilms (Supplementary Fig. 5a, b–d(i)), suggesting a critical role of cellulose and structural proteins in the integrity of mycobacterial biofilms. We also confirmed the enzymatic activity of all the enzymes at the end of the assay to ensure that the activity was not inhibited due to traces of DTT left after washing of the biofilms (Supplementary Fig. 6a–f). We also confirmed whether these enzymes could inhibit the formation of biofilms, 3 h post-exposure to DTT, these enzymes were added to the Mav,

Mab and Mfo cultures. We did not observe biofilm formation in the presence of Cellulase and Proteinase K (Supplementary Fig. 5a, b–d(ii)). To validate the presence of cellulose in the EPS of biofilm matrices of Mav, Mab and Mfo, we isolated cellulose from the pellicle biofilms, DTT induced biofilms and macrocolonies of each of the strains using the Updegraff method[26] and performed FTIR analysis. Peaks in the cellulose isolated from Mav, Mab, and Mfo biofilms corresponded to those obtained from the commercially available cellulose (Fig. 3a–c and Supplementary Fig. 7). These observations suggest that cellulose is present in the different types of the biofilms of NTM, and it plays a crucial structural role in the mycobacterial biofilms. Based on these observations, we hypothesized that cellulose could be used as a biomarker to detect mycobacterial biofilms.

**Mtb forms biofilms inside in vitro granulomas**. Having found cellulose in the biofilm matrices of NTM, we examined whether it could be used as a biomarker to detect mycobacterial biofilms. Since granulomas are the hallmark of Mtb infection, we explored whether Mtb biofilms could be detected in the artificial granulomas. Towards this, peripheral blood mononuclear cells (PBMCs) isolated from human blood were placed in a collagen matrix and infected with Mtb for seven days as described earlier[27]. Typical TB granuloma features were visualized by

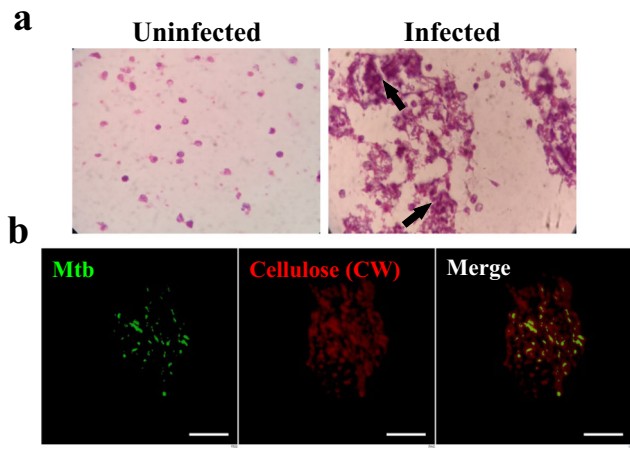

**Fig. 4 Mtb forms biofilms inside the in vitro granulomas. a** Histopathology staining of 5 μm thick sections of uninfected and Mtb infected PBMCs placed on a collagen based 3D matrix visualized on a compound microscope. **b** CW and Auramine staining of the granuloma sections suggesting encapsulation of Mtb bacilli within a cellulose-rich extracellular matrix. All data are representative of three independent biological experiments. Scale bars correspond to 10 μm.

Hematoxylin and Eosin (H & E) staining (Fig. 4a). To detect the presence of Mtb biofilms, deparaffinized sections of in vitro granuloma were stained with CW and analyzed using confocal laser scanning microscopy (CLSM). CW is a fluorescent stain with high specificity towards polysaccharides containing $\beta(1 \rightarrow 4)$ and $\beta(1 \rightarrow 3)$ glycosidic bonds[28]. Mtb cells were visualized through staining with Phenolic Auramine O-Rhodamine B dye. We observed that CW stained the extracellular material surrounding Mtb cells inside the granuloma (Fig. 4b), suggesting that mycobacterial cells are encased within the cellulose matrix. These observations indicate that Mtb forms cellulose wrapped biofilms inside the artificial granulomas.

**Mtb forms biofilms in infected mouse lungs**. Given that the mouse model is often utilized to understand TB pathogenesis and that cellulose is present in an in vitro Mtb granuloma, we next analyzed whether Mtb forms biofilms in mice's lung tissues. Towards this, adult C57BL/6 J mice were infected with Mtb H37Rv. Mtb infected animals and their respective uninfected control animals were sacrificed, and their lungs were aseptically isolated. CFU estimation was utilized to ensure the establishment of Mtb infection. The lung sections were histologically evaluated, and we noticed progressive pathology and striking differences between the infected and the uninfected lungs (Fig. 5a). The presence of cellulose was analyzed through staining with CW. Uninfected mouse lung sections were taken as the negative control. The mycolic acid-containing bacilli were stained with Phenolic Auramine O–Rhodamine B dye. As expected, Mtb cells were visualized in the infected animals. Interestingly, intense CW staining was seen around the Mtb cells (Fig. 5b, c). Fields lacking Mtb cells did not show any presence of CW staining (Supplementary Fig. 8). CW staining was mostly absent in the lung tissues of uninfected animals. To ensure that CW was specifically staining cellulose, lung tissue blocks were treated with Cellulysin Cellulase and then stained with CW. Cellulysin Cellulase is known to degrade cellulose specifically. Expectedly, CW staining was significantly reduced in tissue sections treated with Cellulysin Cellulase, suggesting highly specific staining with CW (Fig. 5b, c). To confirm the presence of cellulose encased Mtb communities inside the mouse lungs, we utilized CBD-mCherry (CBD-mCh) probe.

In this probe, the cellulose-binding domain (CBD) of CenA of *Cellulomonas fimi*[29] is fused with the fluorescent protein mCherry through an octa-glycine linker (Fig. 5d). Since the CBD has a high affinity for cellulose, this novel probe binds cellulose with high affinity. Both the uninfected and infected lung sections were stained with CBD-mCh and analyzed using CLSM. The infected sections showed fluorescence from mCherry that was absent in the uninfected lung tissues (Fig. 5e, f). The cellulose staining with CBD-mCh was reduced tremendously after treatment of the tissue block with Cellulysin Cellulase. These results were again suggestive of the presence of cellulose encased Mtb biofilms in mouse lungs. To confirm cellulose's presence in lung tissues of mice infected with Mtb, the infected and uninfected lung blocks were treated with Cellulysin Cellulase. The quantity of glucose released upon degradation of cellulose was quantitated through measurement of the reduction of 3, 5-dinitrosalicylic acid (DNS). The infected lung blocks from Mtb infected mice led to a significantly higher reduction of DNS upon cellulase treatment compared to lung tissue blocks from uninfected lungs (Fig. 5g). These results point towards the presence of cellulose in lung tissue of Mtb infected mice. These findings were further validated by Raman microscopy of both infected and uninfected lungs. Commercially available cellulose was used as a positive control (Supplementary Fig. 9). The Raman spectral peaks in the infected lung section corresponded to cellulose[30], whereas tissue sections from uninfected mice lacked peaks specific for cellulose (Fig. 5h). These observations confirmed the presence of cellulose in lung tissue from Mtb infected mice.

**Mtb forms biofilms in infected macaque lungs**. Non-human primates are the most relevant model for human TB[31]. We further analyzed whether Mtb forms biofilms in the pulmonary tissue of non-human primates using lung tissue sections from infected and uninfected rhesus macaques. The lung sections were histologically evaluated, and we found remarkable histological differences between the infected and the uninfected lungs (Fig. 6a). We utilized CW staining of lung tissue sections from Mtb infected and non-infected rhesus macaques for exploring the possibility of Mtb biofilms in vivo. The lung sections from uninfected macaques did not stain for the presence of $\beta(1 \rightarrow 4)$ linked D-glucopyranosyl units, whereas lung sections from the infected ones were strongly stained for CW surrounding the Mtb cells (Fig. 6b, c). Importantly, fields lacking Mtb cells in the Mtb infected lung sections also did not stain for cellulose (Supplementary Fig. 10). These observations suggest the presence of Mtb biofilms in non-human primates. Tissue sections were also treated with Cellulysin Cellulase and then stained with CW to rule out non-specific staining of CW. We observed that the fluorescence of CW staining in the Mtb infected lung sections disappeared upon Cellulysin Cellulase treatment suggesting highly specific binding of CW to the cellulose (Fig. 6b, c). These results indicate the presence of Mtb biofilms in the rhesus macaque lung samples infected with Mtb. To confirm cellulose-encased Mtb communities inside the lungs, we also utilized the CBD-mCh probe. Both the uninfected and infected lung sections were stained with CBD-mCh and analyzed using CLSM. Infected macaque lung sections stained for CBD-mCh, whereas lung sections from the uninfected ones failed to do so (Fig. 6d, e). CBD-mCh staining of infected lungs disappeared upon treatment with Cellulysin Cellulase (Fig. 6d, e). The presence of cellulose was further verified by estimating the amount of reducing sugar released upon treating the lung blocks with Cellulysin Cellulase. We observed the release of an appreciable amount of glucose from the infected macaque lungs' tissue blocks upon Cellulysin Cellulase treatment. In contrast, the tissue blocks from uninfected

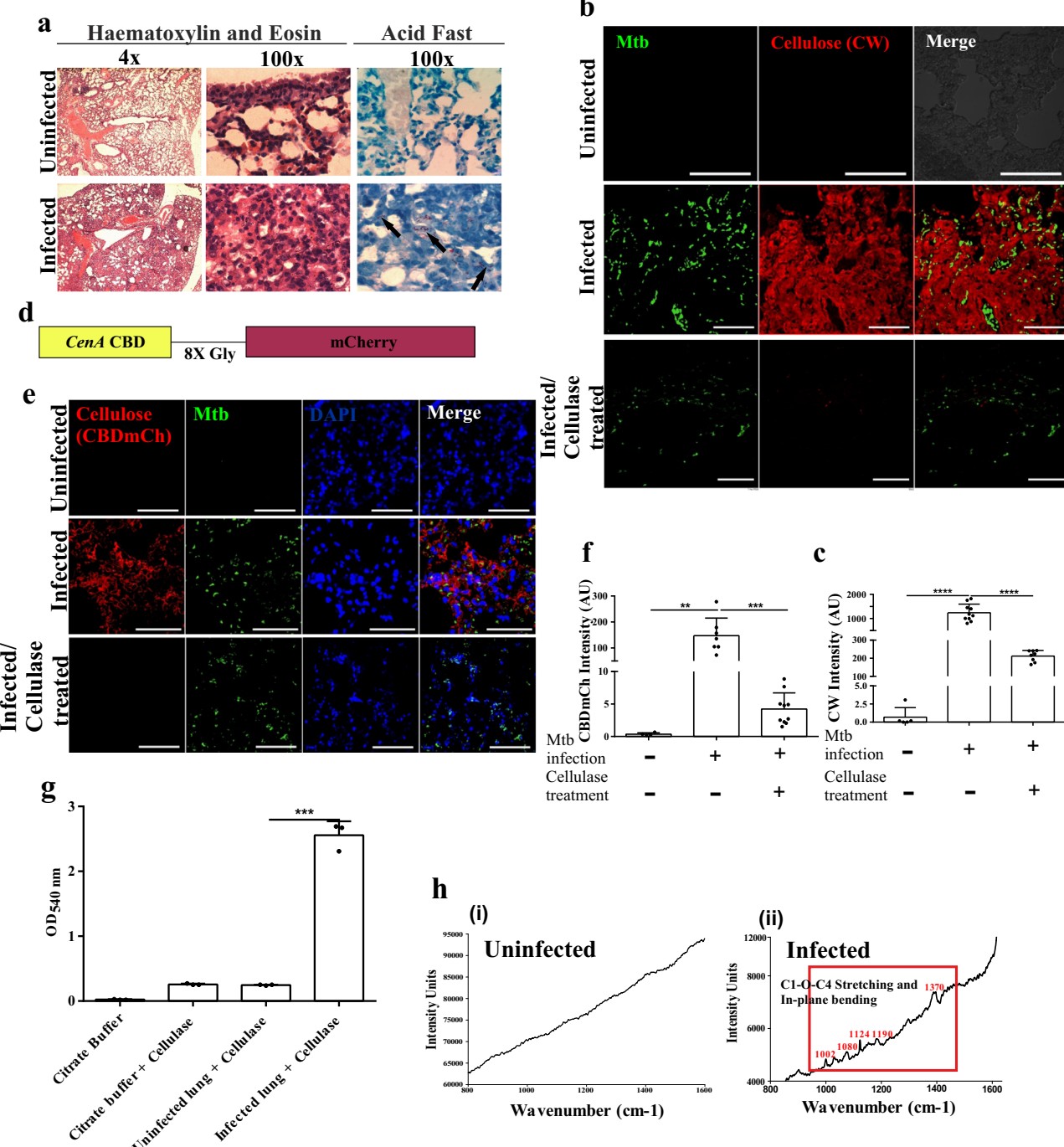

**Fig. 5 Mtb forms biofilms inside the mice lungs. a** Histopathology of granulomatous lesions in the lungs of Mtb infected mice ($n = 5$) in contrast to uninfected ones, as indicated by H & E staining. Clusters of acid-fast bacilli are denoted by arrows. **b**, **c** CW staining and corresponding quantitation (using NIS elements) of Mtb infected, uninfected and Cellulysin Cellulase treated lung sections of mice showing Mtb bacilli (stained with Auramine B–Rhodamine O) encased within a polysaccharide-rich matrix. **d** The schematic construct of the CBD-mCh probe wherein the Cellulose Binding Domain (CBD) of CenA is fused to fluorescent probe mCherry through an octa-glycine linker. **e**, **f** CBD-mCh staining and quantification (using NIS elements) for the experiment described above. **g** DNS assay on uninfected and Mtb infected mice lungs after treatment with cellulase to check the yield of reducing sugar glucose. **h** Raman microscopy of (**i**) uninfected and (**ii**) Mtb infected mice lung sections showing cellulose specific peaks. The column bar graphs presented in figures **c**, **f** and **g** have been plotted in GraphPad Prism 6 and represented as mean (±s.e.m). Statistical significance was determined using Student's $t$-test (two tailed). For **c**, ****$P < 0.0001$, for **f**, **$P = 0.0022$ and ***$P < 0.0001$, and for **g**, ***$P < 0.0001$. All data are representative of three independent biological experiments performed in triplicates. Scale bars correspond to 50 μm. All source data are provided as a Source Data file.

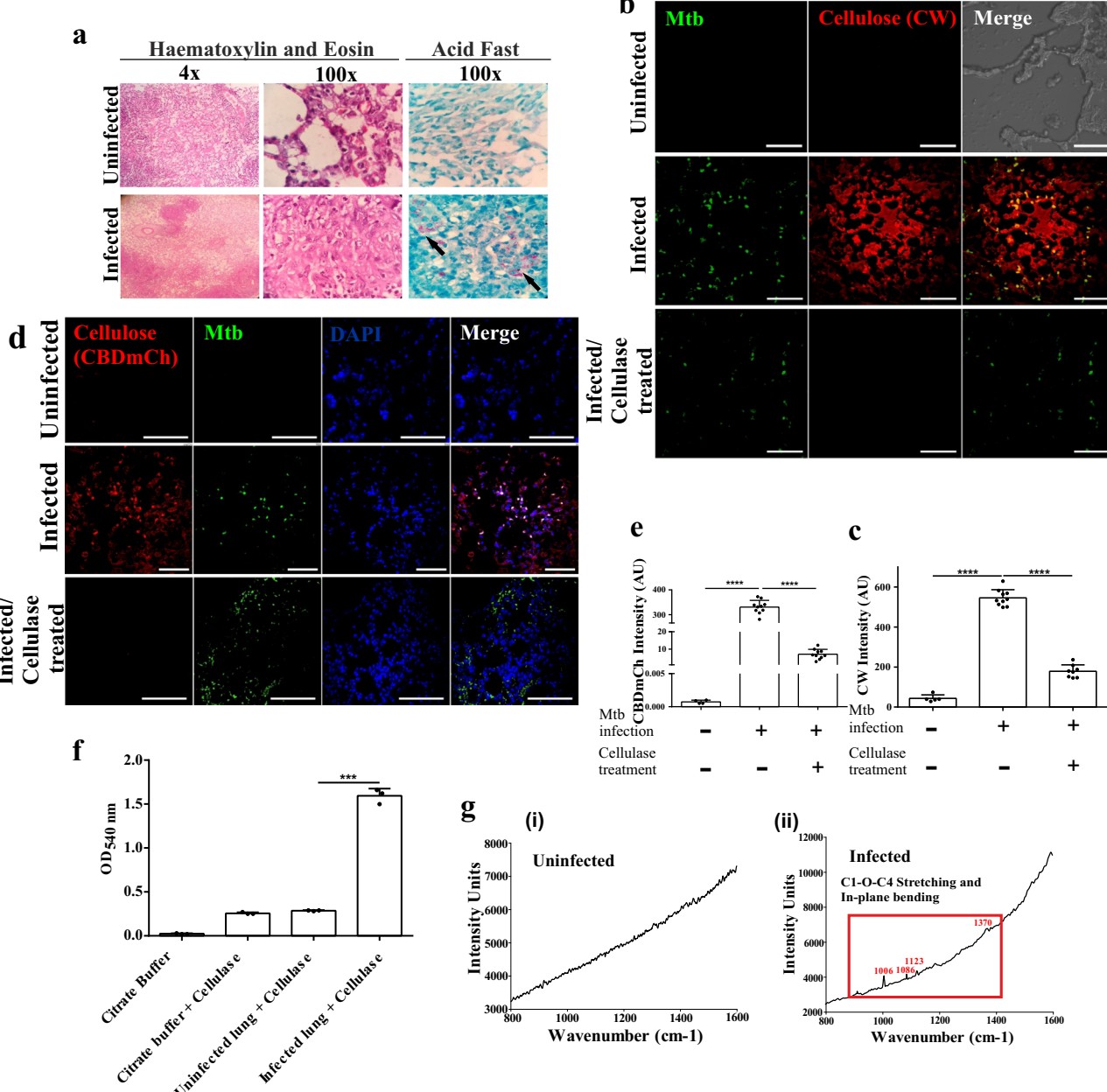

**Fig. 6 Mtb forms biofilms inside the lungs of non-human primates. a** Histopathology of granulomatous lesions in the lungs of Mtb infected rhesus macaques in contrast to uninfected ones, as indicated by H & E staining. Clusters of acid-fast bacilli are indicated by arrows. **b**, **c** Cellulose staining with CW and corresponding quantitation (using NIS elements) of Mtb infected, uninfected and Cellulysin cellulase treated lung sections of rhesus macaques showing Mtb bacilli stained with Auramine B-Rhodamine O. **d**, **e** CBD-mCh staining for cellulose and subsequent quantification (using NIS elements) of Mtb infected, uninfected and Cellulysin cellulase treated lung sections of rhesus macaques showing Mtb bacilli (stained with Auramine B-Rhodamine O). **f** DNS assay of cellulase treated lung tissue blocks from uninfected and Mtb infected rhesus macaque. **g** Raman microscopy of (**i**) uninfected and (**ii**) Mtb infected rhesus macaque lung sections showing cellulose specific peaks in the infected lungs. The column bar graphs presented in figures **c**, **e** and **f** have been plotted in GraphPad Prism 6 and represented as mean (±s.e.m). Statistical significance was determined using Student's $t$-test (two tailed). For **c** ****$P < 0.0001$, for **e** ****$P < 0.0001$ and for **f** ***$P < 0.0001$. All data are representative of three independent biological experiments performed in triplicates. Scale bars correspond to 50 µm. All source data are provided as a Source Data file.

macaque lungs did not release substantial glucose levels on cellulase treatment (Fig. 6f). The presence of cellulose in the Mtb infected lung tissue was also validated using Raman microscopy of lung sections from both infected and uninfected lungs. The Raman spectral peaks for the infected lung section corresponded to those of commercial cellulose, further verifying the presence of cellulose after infecting non-human primates with Mtb H37Rv (Fig. 6g). In summary, these experiments suggest the presence of

Mtb cell masses encased in a cellulose-rich extracellular matrix in the lungs of the infected non-human primates.

**Mtb forms biofilms in infected human lungs**. Having found cellulose in the infected mice and non-human primate lungs, we next analyzed the presence of Mtb biofilms in human lung samples having TB. Towards this, human autopsied lung samples

showing histological features of TB were obtained from the Postgraduate Institute of Medical Education and Research, Chandigarh, India. Similar lung sections from autopsied human cadavers suffering from lung infections other than TB were also collected and used as control. Mtb infection was confirmed using acid-fast staining (Fig. 7a) and Mtb species-specific probe in fluorescence in situ hybridization (FISH) assay, as described earlier[32] (Fig. 7b, Supplementary Fig. 11). These sections were stained with CW and analyzed using CLSM. Consistent with the results from rhesus macaques, the Mtb negative human lung sections did not stain for CW, whereas the Mtb positive lung sections stained for CW (Fig. 7c, d). Mtb infected lungs possess several loci wherein the bacterial number is high, while at others, the number of Mtb cells is less, and at many loci, Mtb cells are absent. Notably, CW staining varied proportionately with the bacillary burden inside the infected lung tissue and almost disappeared in regions with a low or no bacillary load (Supplementary Fig. 12). The CW staining of Mtb positive lung sections disappeared upon treatment with Cellulysin Cellulase (Fig. 7c, d). This phenomenon was further verified using the CBD-mCh probe, which stained the Mtb positive human lung sections and failed to stain the Mtb negative ones (Fig. 7e, f). Upon treatment with Cellulysin Cellulase, infected lungs released much higher glucose levels than the uninfected lungs (Fig. 7g). The presence of cellulose in Mtb positive human lung tissue sections was further confirmed using Raman microscopy of both the Mtb positive and Mtb negative lung sections. In these experiments, microscopy helped in the identification of fields of lung sections having Mtb cells. The Mtb positive lung sections showed spectra matching to cellulose in the fields having Mtb cells. Importantly, fields lacking Mtb cells did not show the Raman spectra similar to that of cellulose. Consistent with these observations, the Mtb negative lung sections did not show characteristic spectra of cellulose (Fig. 7h). Hence, in agreement with data from non-human primates, human lungs also showed the presence of cellulose around the Mtb cells suggesting the presence of Mtb biofilms in humans.

**Biofilms play an important role in protection from the host**. After elucidating that Mtb forms biofilms in mice, macaque, and human lungs, we hypothesized that biofilms protect resident Mtb cells from the host immune system. Towards this, we constructed Mtb strains that independently overexpress and secrete mycobacterial cellulases[21,22,33], Rv0062 (also known as CelA1) and Rv1090 (also known as CelA2b) using Antigen 85b secretion tag[34] (Fig. 8a). These strains grew similar to the vector control (Fig. 8b). They possessed sensitivity to antibiotics, i.e., Rif and INH, and similar metabolic rates as deduced by ATP and NADH levels (Supplementary Fig. 13). The secretion of the cellulases was detected in the culture supernatant (Fig. 8c). Given that cellulose is critical in biofilm formation, these strains were defective in making pellicle biofilms (Fig. 8d). These results agree with an earlier study wherein overexpression of *M. smegmatis* homolog of Rv0062, MSMEG_6752, in *M. smegmatis*, led to diminished ability to grow as pellicle[20]. These strains are also altered in their colony morphology and lack cording phenotype suggesting an essential role of cellulose in the architecture of Mtb colonies (Fig. 8e). Importantly, these strains were also attenuated in biofilm formation in response to DTT (Fig. 8f). Attenuation in the capability was quantifiable using crystal violet assay (Fig. 8f). Mice were independently infected with these strains and their vector control using low dose aerosol infection. Importantly, we found that the engineered strains overexpressing and secreting cellulases (namely, Rv0062 and Rv1090) were unable to cause damage to the lung tissue (Fig. 8h). These changes in the lung pathology were reflected in the lung areas with pathology (Fig. 8i).

These findings were in-line with the fact that the engineered strains were highly attenuated in establishing and maintaining a pulmonary infection (Fig. 8j). Notably, the attenuation (~2 $\log_{10}$ difference in CFU) was witnessed at two weeks post-infection, suggesting that Mtb forms biofilms early during infection, and these biofilms play an essential role in the protection of Mtb cells from the host defenses. Interestingly, CFU of the engineered strains (attenuated for biofilm formation) progressively decreased after four weeks of infection (Fig. 8j), suggesting that biofilms protect the resident Mtb cells from the adaptive immune response as well. This phenotype was more prominent in strains over-expressing Rv0062. These data indicate a critical role of biofilms in the protection of Mtb cells from the innate and adaptive immune system.

**Mtb biofilms protect the resident bacilli from antimycobacterials**. Next, we hypothesized that biofilms could protect the resident bacilli from antimycobacterial agents. Importantly, the currently used chemotherapy regimen for TB was developed using the mouse model of TB[35–37]. Thus, this model is the best model for illustrating the protection offered by biofilm onto the resident bacilli against front-line TB drugs. To test the above-stated hypothesis, mice were infected with Mtb through a low-dose aerosol challenge. One month post-infection, mice were divided into two sets. One set of mice was orally treated with a combination of Rifampicin (Rif) and Isoniazid (INH) while the other group of mice received nebulized Cellulysin Cellulase (or heat-inactivated enzyme as negative control) in addition to the oral treatment with Rif and INH (summarized in study design, depicted in Fig. 9a). Lung histopathology and bacillary load were analyzed at specified time points to ascertain biofilms' role in protection from antimycobacterials. Interestingly, the histopathological analysis suggested that pulmonary tissue destruction was minimal in the group that received cellulase along with the antimycobacterials (Fig. 9b–d). We evaluated whether inhalation of nebulized Cellulysin Cellulase degrades the biofilms in vivo through CBD-mCh staining of the lung sections. CBD-mCh staining suggested a significant reduction in the cellulose stained lung area of the mice that were nebulized with cellulase along with Rif and INH treatment (Fig. 9e, f). CFU analysis revealed a significant decrease in the viable number of bacilli in the mouse lungs after Rif and INH treatment (Fig. 9g). Importantly, chemotherapy-induced clearance of Mtb was significantly enhanced in mice receiving nebulized cellulase. On the other hand, another group of mice nebulized with heat-inactivated cellulase and treated with Rif and INH had a similar bacterial count as those treated with Rif and INH alone (Fig. 9g). Thus, these results suggest that biofilms protect Mtb in vivo against the chemotherapeutic agents. In light of these findings, new treatment methods capable of clearing Mtb biofilms need to be developed. It is tempting to propose that the nebulization of biofilm degrading molecules/enzymes could be used as an adjunct therapy for boosting the action of antimycobacterials.

## Discussion

TB is a chronic disease that displays several key features of a biofilm infection. However, whether Mtb forms biofilms in vivo is not known. We established that cellulose is present in biofilms of fast and slow-growing mycobacterial species and thus could be used as a biomarker of mycobacterial biofilms. Using cellulose as a biomarker of mycobacterial biofilms, we established that Mtb forms biofilms in the lungs of mice, non-human primates, and humans. Furthermore, we have demonstrated that biofilms play a crucial role in the establishment of Mtb infection and protect residing bacilli from immune response and antimycobacterial agents.

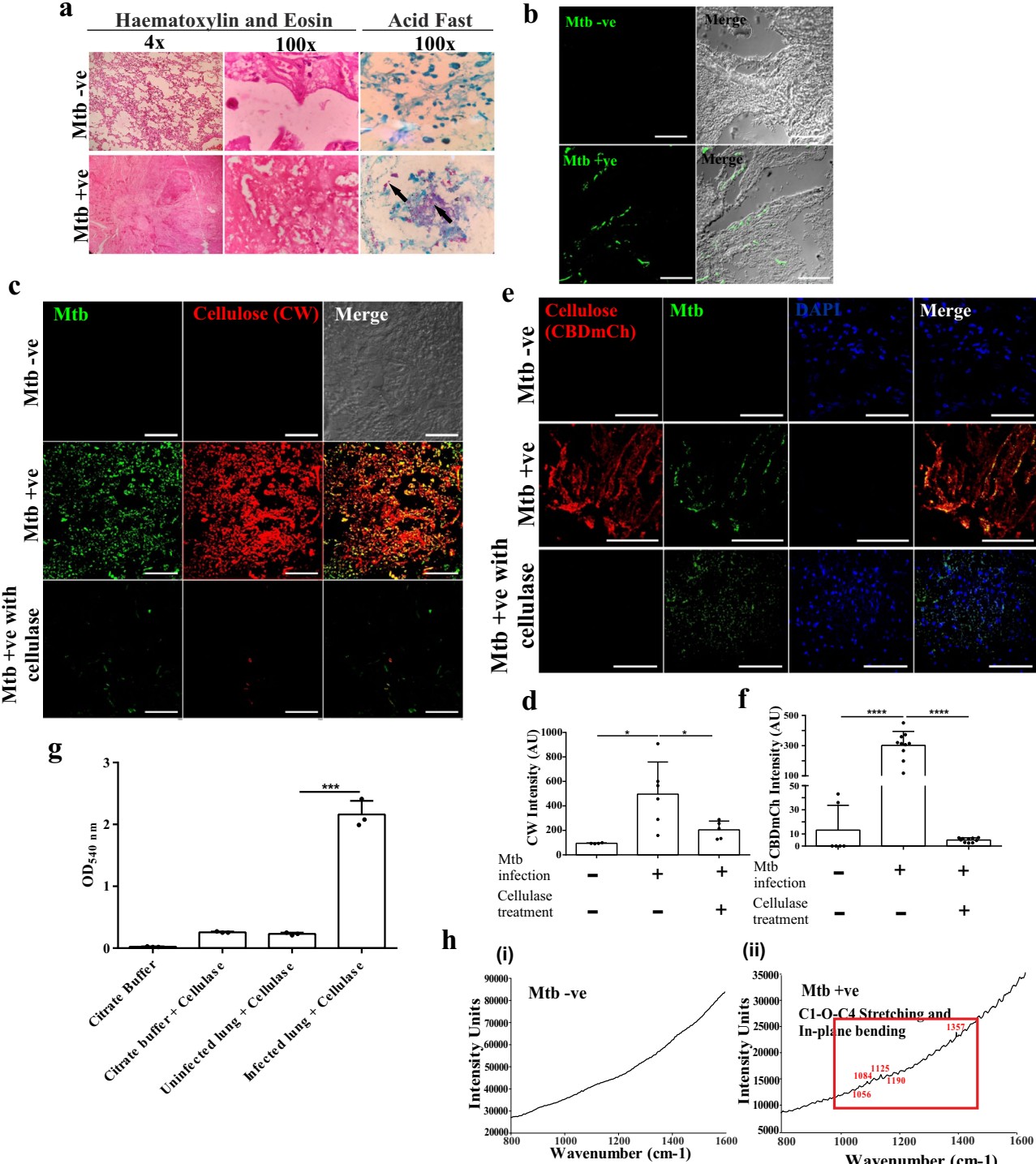

**Fig. 7 Mtb forms biofilms in the human lungs. a** H & E staining of human lung tissue sections. The clusters of acid-fast bacilli are denoted by arrows. **b** FISH with Mtb specific 16 S primers was utilized for identifying lung tissues infected with Mtb. **c, d** Cellulose was visualized using CW staining in Mtb positive and Mtb negative human lung tissue sections. Mtb bacilli were visualized using Auramine B-Rhodamine O staining. Mtb positive tissue sections were also stained with CW after Cellulysin Cellulase treatment. **e, f** CBD-mCh staining and visualization of Mtb (Auramine B-Rhodamine O) in Mtb positive and Mtb negative lung tissue sections. **g** DNS assay on Mtb positive and Mtb negative human lung tissue blocks after treatment with Cellulysin Cellulase. **h** Raman microscopy of (**i**) Mtb positive and (**ii**) Mtb negative human lung sections. The column bar graphs presented in figures **d**, **f** and **g** have been plotted in GraphPad Prism 6 and represented as mean (±s.e.m). Statistical significance was determined using Student's $t$-test (two tailed). *$P < 0.05$, **$P < 0.001$, ***$P < 0.0001$. For **d** *$P = 0.0169$ (column A vs column B) and *$P = 0.0401$ (column B vs column C), for **f** ****$P < 0.0001$ and for **g** ***$P = 0.0001$ All data are representative of three independent biological experiments performed in triplicates. Scale bars correspond to 50 μm. All source data are provided as a Source Data file.

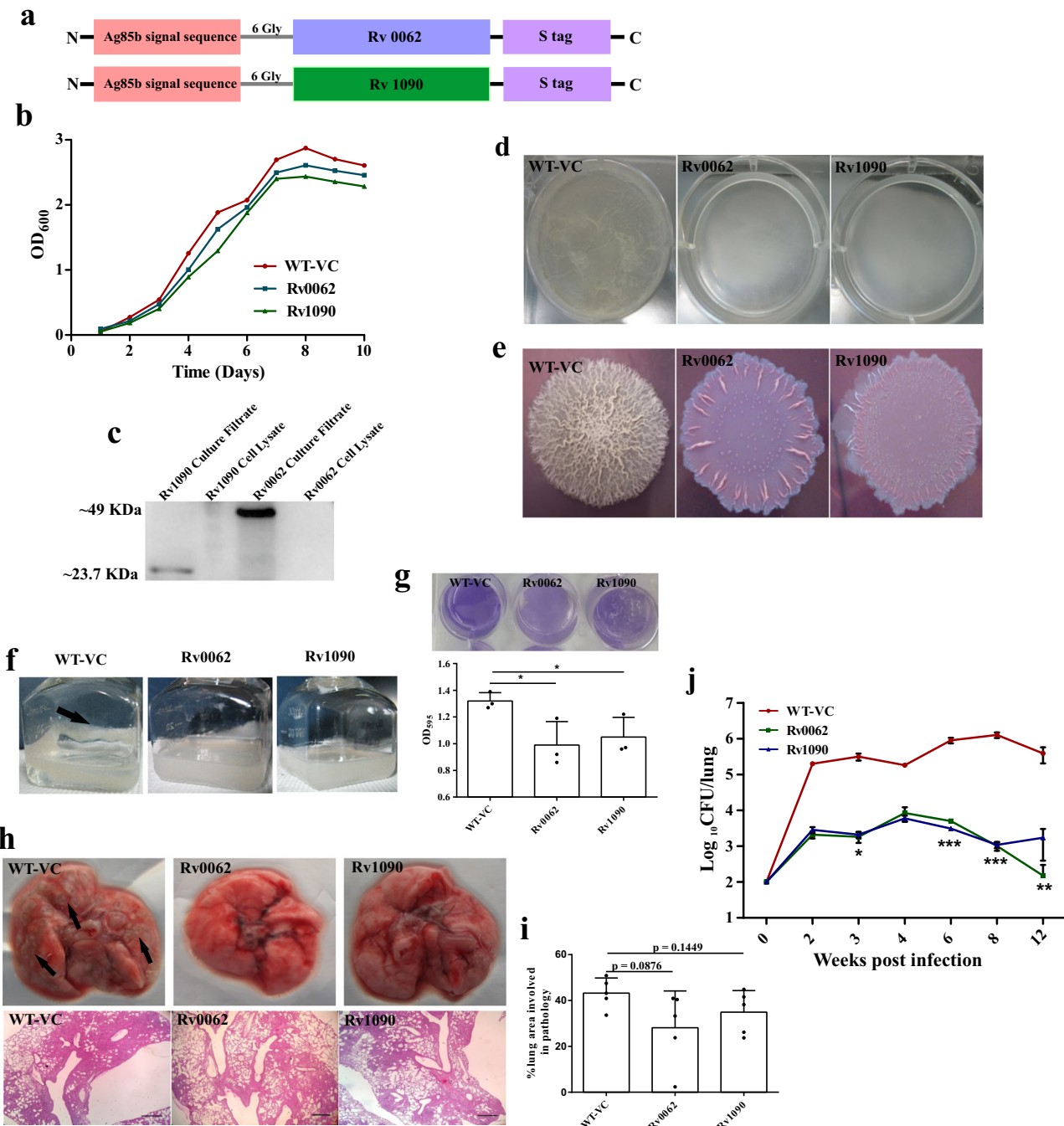

**Fig. 8 Mtb biofilms protect the bacilli from the host immune system. a** Construct design for engineering Mtb strains overexpressing endogenous Mtb cellulases (Rv0062 and Rv1090). **b** Growth profile of planktonic cultures of Mtb strain with vector control or those overexpressing Rv0062 and Rv1090. **c** The culture supernatant of the above-described cultures was used in Western blot analysis. Antibodies specific to S-tag were utilized to detect the secretion of cellulases outside Mtb cells. Uncropped image of the blot is provided in Source Data. **d** Pellicle biofilm profile of WT-VC, Rv0062, and Rv1090, suggesting that cellulase overexpressing strains were severely defective in pellicle formation. **e** Macrocolonies of WT-VC, Rv0062, and Rv1090 on 7H11 agar supplemented with Congo red and Coomassie Brilliant Blue G250 dye. **f, g** Biofilms of WT-VC, Rv0062, and Rv1090 induced by thiol reductive stress using DTT shows defect in the engineered strains, as indicated by CV assay quantitation. The biofilm formed by WT-VC (**f**) has been shown by black arrow. **h** C57BL/6 J mice were independently infected with vector control (WT-VC) or strains overexpressing cellulases Rv0062 and Rv1090 using a low dose of aerosols. Gross lung pathology (upper panel) and histopathology (lower panel) of mice lungs infected with WT-VC, Rv0062, and Rv1090. **i** Lung pathology post-infection with Mtb strains, as mentioned in **h**, was quantitated using ImageJ software. **j** Survival inside mice lungs ($n = 5$) was estimated using CFU analysis. The data presented in figures **g**, **i** and **j** have been plotted in GraphPad Prism 6 and represented as mean (±s.e.m). Statistical significance of **j** was determined using two way ANOVA. *$P < 0.05$, **$P < 0.01$, ***$P < 0.001$. And the statistical significance of **g** and **i** was determined using Student's $t$-test (two tailed). For **g** *$P = 0.0375$ (column A vs column B) and *$P = 0.0431$ (column A and column C). All data are representative of three independent biological experiments performed in triplicates unless otherwise mentioned. Scale bars correspond to 200 μm. All source data are provided as a Source Data file.

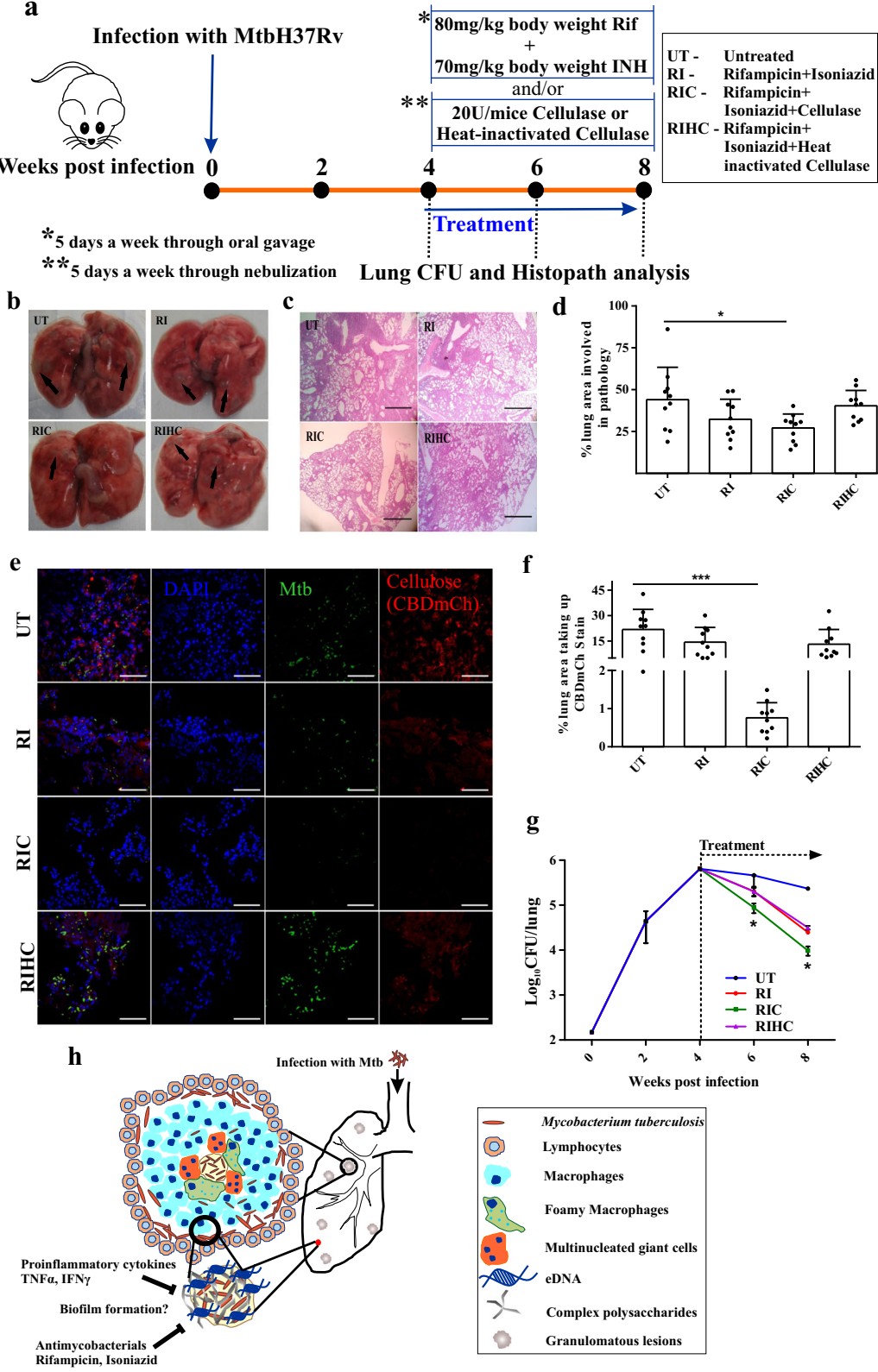

An essential finding of this study was the identification of cellulose as a critical structural component of the monospecies biofilms of Mav, Mab and Mfo. Earlier studies have identified extracellular DNA as a component of Mav[38], Mab and Mfo biofilms[39]. In this study, we observed that slow and fast-growing mycobacteria use cellulose as a component of EPS. Furthermore, the degradation of cellulose through enzymes led to the disruption of biofilms. This disruption was higher than observed with DNase and thus suggested a more significant role of cellulose as a structural component.

Since this study indicates that cellulose is used as a structural component of mycobacterial biofilms, and since cellulose is absent in humans, we utilized it as a biomarker to detect the presence of Mtb biofilms in vivo. We discovered the presence of

**Fig. 9 Mycobacterial biofilms protect the resident bacilli from antimycobacterial agents. a** Schematic representation of Mtb infection followed by drug treatment regime in mice ($n = 4$, an experiment performed twice). **b–d** Gross lung pathology, histopathology analysis, and pathology scoring of mice lungs infected with Mtb, followed by treatment. **e, f** CBD-mCh staining and quantification (using NIS elements) in lung sections of Mtb infected mice subsequently treated as described in **a. g** Survival of Mtb inside mice 2 and 4 weeks post-treatment with Rif and INH orally and/or cellulase or heat-inactivated cellulase through nebulization. **h** Schematic representation of Mtb biofilms inside the human lungs. Infection with Mtb leads to the formation of granulomatous lesions in the human lungs. A granuloma is a compact immunological structure predominated by macrophages at the center, which, in turn, differentiate into other cell types like foamy macrophages and multinucleate giant cells having lipid droplets, in a specialized manner. The periphery of the granuloma is comprised of T and B lymphocytes. However, in our proposition, the necrotic core of the granuloma, consisting of Mtb bacilli is structured as a biofilm, which is composed of extracellular matrix including cellulose as one of the major components, can act as a barrier to both antimycobacterial agents as well as a host defense mechanism, thus rendering protection to the bacteria. The data presented in figures **d, f** and **g** have been plotted in GraphPad Prism 6 and represented as mean (±s.e.m). Statistical significance of **g** was determined using two way ANOVA, *$P < 0.05$ and statistical significance of **d** and **f** were determined using Student's $t$-test (two tailed). *$P < 0.05$, ***$P < 0.0001$. For **d** *$P = 0.0205$ and for **f** ***$P < 0.0001$. All data are representative of three independent biological experiments performed in triplicates unless otherwise mentioned. Scale bars correspond to 200 µm. All source data are provided as a Source Data file.

Mtb biofilms in the lungs of mice, non-human primates, and humans. We observed several clusters of mycobacterial cells encased in a cellulose matrix. Given that the presence of biofilms in vivo is debatable, we utilized several complementary biochemical methods for the detection of cellulose around the mycobacterial cluster. However, it must be noted that all these methods rely on detecting β-1,4-glycosidic linkage of the glucose anomers that may be present in other polysaccharides as well. Further characterization through purification of cellulose (or cellulose-like material) from the lung tissue followed by analysis with methods like solid-state NMR, Thermogravimetric analysis, X-ray powder diffraction, etc., would be more definitive and could even through light on different allomorphs of cellulose in the in vivo biofilms. Isolation of cellulose in large quantities from tissue samples for analysis using the techniques mentioned above is technically challenging and is beyond the scope of this study. Earlier studies have detected an electron transparent zone (ETZ) around the intracellular mycobacterial cells[40,41]. However, the biochemical nature of the ETZ has remained elusive. It is plausible that compactly packed cellulose or other polysaccharides of EPS could be the components of ETZ. Further studies are required to test this hypothesis. Nevertheless, these observations provide a new perspective to our understanding of TB pathogenesis. The presence of Mtb biofilms in pulmonary tissues aligns with the chronic nature of TB. Also, it provides a plausible explanation for the evasion of the immune system by Mtb and the phenotypic drug tolerance observed in vivo. This discovery challenges the current dogma that focuses on Mtb as an intracellular pathogen[42,43]. However, the intracellular pathogen doctrine has overlooked the presence of extracellular Mtb in lung tissues. Recent studies suggest the presence of extracellular Mtb clusters that could survive treatment with antimycobacterials[44,45]. Our findings indicate that such extracellular clusters may represent Mtb biofilms. We believe that Mtb biofilms could also be present at sites other than granulomas, wherein Mtb cells could hide away from the immune system. We believe that there are at least four distinct populations of Mtb in the pulmonary tissues. A fraction of the bacterial population resides inside macrophages where they thrive and could lead to activation of the immune system and manifestation of the disease. The activated immune system could derive the formation of persisters inside macrophages[46], representing the second distinct population. A third population of extracellular bacilli may survive in the necrotic core of granulomas and exhibit a persister like phenotype. A fourth population consists of extracellular Mtb cells that organize themselves into biofilms. Persisters residing intracellularly along with extracellular biofilms as well as at the necrotic core contribute to treatment failure and recurrence of infection (Fig. 9h). Eradication of these populations is critical to treat TB effectively.

During the development of the TB treatment regimen, the primary focus has been to kill the actively replicating cells in mouse model[35–37]. Lately, the focus has shifted to tackling persisters as well[47,48]. In our view, new screens should be developed by identifying compounds capable of killing mycobacterial cells inside the biofilms.

Treatment of TB primarily relies upon antimycobacterial agents. Theoretically, molecules disrupting biofilms could aid in the killing of the pathogen by the antibacterial agents. A classic example of this is the use of DNase as an adjunct therapy for cystic fibrosis[49]. A close association of cystic fibrosis and *Pseudomonas aeruginosa* infection in the lungs has been established[50]. Importantly, literature has established that the capability of *P. aeruginosa* to form biofilms in the lungs is critical for the manifestation of cystic fibrosis[51]. DNA is an integral part of the *P. aeruginosa* biofilms, and its disintegration leads to disruption of biofilms[52]. The utilization of DNase as an adjunct therapy for cystic fibrosis works in two ways. Firstly, it leads to the degradation of the host generated extracellular DNA in the mucus leading to its thinning and easing up the air flow[53]. Second, its usage could lead to disruption of *Pseudomonas* biofilms and may facilitate the killing of the pathogen by antibacterial agents[54]. In the light of these, and the observation that the administration of nebulized cellulase helps in amelioration of infection, it is tempting to draw a parallel of TB treatment with cellulase with the FDA approved usage of DNase as an adjunct therapy for cystic fibrosis. However, further studies are required for establishing a clear utility of cellulase as an adjunct therapy in the treatment of TB.

A synergistic regimen capable of killing actively replicating cells, persisters residing in macrophages, and biofilm residents could shorten the treatment duration and decrease relapse rates of TB. As a proof of concept for this approach, we have demonstrated that the nebulization of cellulase (that disrupts biofilms but does not kill Mtb cells) could enhance the killing by the combination of INH and Rif. These findings agree with an earlier study wherein an inhibitor of biofilm formation showed a synergistic effect with INH and Rif controlling Mtb infection in mice[14]. However, these are preliminary studies but could pave the way for the development of cellulase nebulization as an adjunct therapy for TB that could potentiate antimycobacterial agents through disrupting EPS in the mycobacterial biofilms.

Another important finding of this study was that biofilms are critical for the establishment of mycobacterial infection. To demonstrate biofilms' role in protection from the immune system, we engineered strains defective in biofilm formation through overexpression and secretion of cellulase, namely Rv0062 and Rv1090. These strains have a similar metabolic profile, growth rate, and drug sensitivity profile as that of vector controls. Still,

these strains are attenuated for pellicle formation and DTT induced biofilm formation. Their colony morphologies are strikingly different than of the Mtb, and they lack cording. These observations are in agreement with an earlier study, demonstrating that the overexpression of Rv0062 homolog in *M. smegmatis* renders attenuation in biofilm formation[20]. We expected attenuation of these strains for survival in mice, but we were surprised by the degree of attenuation in the second week. These findings suggest that biofilm formation at the early stages helps Mtb to establish the infection. Notably, the survival of biofilm defective strains reduced progressively after the onset of the adaptive immune system suggesting that biofilms play an essential role in the protection of Mtb from the immune system. However, this study has not explored biofilms' contribution to the innate and adaptive immune response's evasion. Such a detailed analysis is beyond the scope of this study. Furthermore, we believe that since cellulose is now established as a biomarker of mycobacterial biofilms, it could also be used to analyze whether extrapulmonary TB cases are also biofilm infections. Furthermore, it would be useful to detect the presence of NTM biofilms in vivo wherever relevant.

Although we have established that in vivo biofilms could be disrupted through the administration of nebulized cellulase and thereby enhancing the mycobacterial killing by antimycobacterial agents, however, it must be noted that cellulase being a foreign protein to mammals, could elicit an immunological response in the recipient. Prolonged exposure to cellulase under industrial settings is known to elicit Asthma[55], Pharyngeal edema, and Rhinitis[56]. Systematic studies on the immunological response and toxic effects of cellulase administration need to be undertaken to explore further the possibility of using cellulase as an adjunct therapy. In an ideal situation, small molecules capable of disrupting Mtb biofilm in vivo would be preferred as an adjunct therapy. Such molecules could be discovered through high throughput screens utilizing the disruption of biofilms as a readout. It will need more work than described in this manuscript. Furthermore, we would like to point out that extrapulmonary TB is more difficult to treat, and Mtb biofilms are likely to be the underlying reason. More research is required to analyze the role of Mtb biofilms in the pathogenesis of extrapulmonary TB.

## Methods

**Bacterial strains and growth conditions**. *Mycobacterium avium* 2285 (NR-44265) (Mav), *Mycobacterium abscessus* (NR-44261) (Mab) and *Mycobacterium fortuitum* (ATCC®6841) (Mfo) were cultured in Middlebrook 7H9 broth supplemented with 5% OADC (Oleic Acid-Albumin-Dextrose-Catalase), 0.2% Glycerol and 0.1% Tween 80. Liquid cultures were aerated with shaking at 200 r.p.m. at 37 °C on an Innova 42 shaker incubator. All experiments with Mav, Mab, and Mfo were performed in the BSL-2 facility of CSIR-IMTech according to the institutional biosafety guidelines. Planktonic cultures of Mtb H37Rv (ATCC 27294) were grown at 37 °C under aerobic conditions in Middlebrook 7H9 broth supplemented with 0.2% glycerol, 0.1% Tween 80 and 10% OADC. Wherever necessary, bacteria were grown on solid Middlebrook 7H11 agar plates supplemented with 0.5% glycerol and 10% OADC. The cellulase overexpressing strains of Mtb H37Rv were grown on Middlebrook 7H9 broth or solid Middlebrook 7H11 agar along with additional Kanamycin (25 µg/ml). For growth curve experiments, exponentially growing cultures of the engineered strains of Mtb were washed and inoculated in 7H9 medium with all the supplements to achieve an initial $OD_{600}$ of 0.1. Their growth rate was recorded for ten days until they reached the stationary phase of growth. All experiments with virulent Mtb were performed in the BSL-3 facility of CSIR-IMTech according to institutional biosafety guidelines.

**Biofilm assays and colony morphology**. For pellicle biofilm formation, saturated cultures of Mav, Mab, Mfo and engineered Mtb strains along with the vector control were inoculated (10% inoculum) in Sauton's media in 6 well plates and incubated at 37 °C for approximately 20 days without shaking. In order to form submerged biofilms, logarithmic-phase cultures of Mav, Mab and Mfo were induced with either 6 mM of βME, or oxidized DTT, or reduced DTT for 29 hr at 37 °C. For the engineered Mtb strains, exponential cultures were exposed to 6 mM

reduced DTT for submerged biofilm formation. For colony morphological studies, 5 µl of each culture were spotted on solid Middlebrook 7H11 agar either as is or supplemented with CR (40 µg/ml), CBB (20 µg/ml). The plates were incubated at 37 °C for approximately four weeks.

**Crystal violet assay of biofilms**. The crystal violet assay of Mav, Mab, and Mfo biofilms was performed as described earlier[17]. Briefly, the crystal violet assay was done in 12 or 24-well plates, as required. After the formation of the bacterial biofilm, the media above the biofilm surface was removed, and 1 ml of 1% crystal violet was added to the biofilm. It was incubated for 10 min at 37 °C. The stain was removed, and the biofilm was gently washed twice with 1× PBS. The bound crystal violet was then extracted by a 10 min incubation at 37 °C with 1 ml of 95% ethanol. The absorbance of extracted crystal violet was measured at 600 nm on a spectrophotometer.

**DTNB assay of the bacterial biofilms**. 5, 5′-Dithio-bis-(2-nitrobenzoic acid) (DTNB) quantitates free thiol groups in a solution. DTNB reacts with free thiols to yield disulfides and NTB (2-nitro-5-thiobenzoic acid), a yellow-colored compound whose absorption at 412 nm gives a direct measure of the level of intracellular thiols. Briefly, cells of Mav, Mab and Mfo were exposed to 6 mM concentration of each of βME, DTT oxidized and DTT reduced. 29 h post-exposure, cells were washed thrice with 1× PBS and finally lysed in a bead beater using lysing matrix B. The supernatants containing the free thiols were collected after centrifugation, and to 50 µl of the sample, 950 µl of 0.1 mM of DTNB (Sigma) solution was added. After 2 min incubation at room temperature, each of the samples' absorbance was measured at 412 nm.

**Drug susceptibility testing**. Submerged biofilms of Mav, Mab and Mfo mediated by thiol reductive stress were subjected to Bedaquiline (0.01 µg/ml for Mav, 0.25 µg/ml for Mab and 0.125 µg/ml for Mfo) at 1×, 10× and 100× MIC and incubated for 48 h at 37 °C. The biofilms were ruptured by 5 mg/ml Cellulase for 3 h, serially diluted in 1× PBS, and plated on 7H11 agar plates.

**Enzyme inhibition assay**. Thiol reductive stress mediated submerged biofilms of Mav, Mab, and Mfo were treated with different enzymes, either 3 or 29 h after induction with 6 mM reduced DTT. The biofilms were treated with Cellulase (*T. viridae*, Calbiochem) at 5 mg/ml concentration in 50 mM citrate buffer, Turbo DNase (0.8 U/ml) (Invitrogen), Proteinase K (*T. album*, Sigma) at a concentration 0.1 mg/ml, Lipase (*C. viscosum*, Calbiochem) at a concentration of 0.5 mg/ml in 100 mM Potassium phosphate buffer (pH 8.0) containing 1 mg/ml BSA, and α-amylase (*B. licheniformis*, Sigma) at a concentration of 500 U/ml. Proteinase K treatment was given for 9 h, whereas Cellulase, Lipase, DNase treatment for 6 h, and α-amylase treatment for 3 h.

**Enzyme activity in the biofilms**. DNaseI activity was checked using 1 mg/ml Salmon sperm DNA in acetate buffer, which was incubated with a range of enzyme units for 1 h at 37 °C. The standard curve was plotted after taking the absorbance reading at 520 nm. Salmon sperm DNA was also incubated with biofilm supernatant, and the $OD_{520}$ reading was measured. Based on the standard curve, the DNaseI activity was checked. Lipase activity was measured using Oleic acid as the substrate, which was incubated with different enzyme concentrations for 1 h at 37 °C. The absorbance was recorded at 570 nm. The absorbance obtained after incubating the substrate with biofilm supernatant was measured from the standard curve. 0.2% of soluble starch solution was used as a substrate for Amylase. After 5 min incubation of the substrate with a range of concentrations of the enzyme, an acidic iodine solution was added. Iodine would bind any starch molecule that had not been degraded by the enzyme and yield a purple color giving an absorbance at 620 nm. The activity of the enzyme in the biofilm supernatant was calculated from the standard curve. The activity of Proteinase K was enumerated using 1 mg/ml BSA as substrate, with 30 min incubation at 37 °C, after which the absorbance was recorded at 595 nm. The enzyme activity in the biofilm supernatant was estimated from the standard curve. Cellulase activity was evaluated using 1 mg/ml Cellulose as substrate using DNS assay as mentioned before. Different concentrations of cellulase were spotted along with biofilm supernatant on Carboxymethyl cellulose (CMC) agar plates and incubated at 27 °C for 16 h. Then the plates were flooded with 0.1% CR for 20 min and washed with 1 M NaCl. The size of the clear zones determines the activity of the enzyme on the cellulose-containing plates.

**Cellulose isolation, quantitation, and FTIR analysis**. The cellulose from the EPS of all the three models of Mav, Mab, and Mfo biofilms was isolated as described earlier[17]. Briefly, the biofilms were treated with Acetic Nitric reagent (150 ml 80% acetic acid + 10 ml concentrated Nitric acid) in a boiling water bath for 30 min. Undissolved EPS that contained cellulose was centrifuged at 12,000 *g* for 30 min and dissolved in Trifluoroacetic acid (TFA) (Sigma). FTIR of the TFA dissolved cellulose was performed in a Bruker VERTEX 70/70 v FT-IR instrument with microcrystalline cellulose (Sigma) dissolved in TFA as control.

**Confocal laser scanning microscopy (CLSM) of biofilm samples**. Mav, Mab, and Mfo biofilms were made by 6 mM DTT treatment in 8 chambered glass slides. These were stained with fluorescent dyes like Texas red (0.5 mg/ml, Molecular Probes), SYPRO Ruby (Molecular Probes), Concanavalin A-Alexa Fluor 647 (200 μg/ml, Molecular Probes) Propidium iodide (15 μM, Sigma). Biofilms were stained with Con A for 45 min, with Texas red and SYPRO Ruby for 20 min and with Propidium iodide for 5 min. Post staining, samples were washed three times with 1X PBS and viewed under CLSM.

**Construction of cellulase overexpressing strains of Mtb H37Rv**. The genes *Rv0062* (GenBank: CCP42784.1) and *Rv1090* (GenBank: CCP4384.1) were cloned in *E. coli-Mycobacterial* integrative shuttle vector pMV761 under the control of the constitutive *hsp60* promoter. These genes were cloned downstream to the signal sequence of mycobacterial *Antigen 85b* attached through a hexa-glycine linker and upstream to an S-tag using cloning services of GenScript, USA. The plasmids were transformed in Mtb H37Rv using standard transformation procedures and grown on 7H11 agar supplemented with 0.5% Glycerol, 10% OADC and Kanamycin (25 μg/ml). The Mtb strain containing the empty pMV761 vector was used as the vector control.

**Immunoblotting**. Immunoblotting was performed using culture filtrates and cell lysates collected from exponentially growing cultures of cellulase overexpressing Mtb strains. For culture filtrates, the cell pellets were removed, and supernatants were filtered through 0.22-micron filters to remove any residual cells. For cell lysates, cells were resuspended in lysis buffer (PBS containing a protease inhibitor cocktail) and lysed in the presence of silica beads (100 μm Lysing matrix B) in a FastPrep at a speed setting of 5.5 for 5–6 cycles of 30 s each. The supernatant was collected by centrifugation and filtered through a low protein binding 0.22 μm membrane filter. For Western blot analyses, around 30 μg protein, estimated using the BCA protein estimation method, from both culture filtrates or cell lysates were resolved by SDS-PAGE and transferred to PVDF membranes (MDI). To identify the S-tagged Rv0062 and Rv1090 proteins, the membranes were probed with anti-S primary antibody (1:2500 dilution) as appropriate and visualized using a horse-radish peroxidase-conjugated secondary antibody (1:10,000 dilution).

**Isolation of PBMCs from human whole blood**. Research work with human blood was performed after seeking approval from the Institutional Ethics Committee of CSIR-IMTech. Human blood was collected from Rotary Blood Bank, Chandigarh, India, from healthy volunteers as per written informed consent. All donors were negative for the standard panel of blood-borne infections. Peripheral Blood Mononuclear Cells (PBMCs) were isolated using the standard protocol of Ficoll-Histopaque density gradient centrifugal separation. Briefly, 15 ml of whole human blood was mixed with an equal volume of 1X DPBS (without $Ca^{++}$ and $Mg^{++}$), and 30 ml of diluted blood was carefully layered using a serological pipette on 15 ml Histopaque-1077 (Sigma) in a 50 ml falcon. The mixture was centrifuged with brakes OFF at 200 $g$ for 20 min, and then carefully, the PBMC layer was aspirated using a pipette. The cells were washed thrice for the removal of any residual Histopaque-1077 and for the removal of platelets. After washing, PBMC aliquots of $5 \times 10^6$ cells per vial were cryopreserved in 10% DMSO and 20% FBS containing RPMI media for extended storage at −150 °C. When needed, PBMC vials were thawed and washed in non-supplemented RPMI containing 10% serum. They were further counted using Trypan Blue dye exclusion method.

**Formation of in vitro granuloma**. An in vitro granuloma model of TB infection was established using previously described protocols[27]. Briefly, an extracellular matrix was prepared by mixing 950 μl Collagen I solution, 50 μl 10× DPBS solution and 10 μl 1 N NaOH, finally adjusted to a pH of 7.0 and kept at 4 °C. PBMCs were mixed with the matrix at $5 \times 10^5$ cells/100 μl/chamber of an eight chambered glass slide at room temperature and allowed to set by incubating at 37 °C incubator with 5% $CO_2$ for 45 min. Cells were then infected with Mtb H37Rv at an MOI of 0.1, considering the presence of macrophages in PBMCs to be 5%. Samples were incubated at 37 °C incubator with 5% $CO_2$, and the media was changed on day four and again on day 7. After seven days of incubation, cells were fixed with 4% PFA overnight and then carefully removed intact from the chambered glass slide for paraffin embedding and sectioning and histological staining.

**Mouse tuberculosis infection model and ethics statements**. All experiments pertaining to mice were approved by the Institutional Animal Ethics Committee of CSIR-IMTech (Approval no IAEC/17/27). All the mice experiments were performed according to the guidelines issued by the Committee for the Purpose of Control and Supervision of Experiments on Animals (No.55/1999/CPCSEA) under the Prevention of Cruelty to Animals Act 1960 and amendments introduced in 1982 by the Ministry of Environment and Forest, Govt. of India. Mice were maintained and bred in the animal house facility of CSIR-IMTech. Animal infections and subsequent studies were performed in the BSL-3 facility of CSIR-IMTECH, as described earlier[57]. Briefly, eight to ten weeks old C57BL/6 J mice were infected through the aerosol route with passaged Mtb H37Rv or the engineered strains of Mtb along with the vector control using a nebulizer having the parameters of 15 min preheat cycle, 45 min nebulization cycle, 30 min cloud decay

cycle followed by 15 min decontamination by an aerosol inhalation exposure system (Glas-Col, USA). The actual bacterial load delivered to the animals was enumerated from three aerosol-challenged mice, one-day post aerosol challenge. The animals were found to achieve a bacillary deposition of 100 to 150 CFU in the lungs. The animals were sacrificed at specific time points. Lungs were isolated aseptically from the euthanized animals, homogenized in sterile 1× PBS and plated after serially diluting the lysate on 7H11 agar plates, supplemented with 10% OADC and antibiotics [50 μg/ml Carbenicillin in water (Gold Biotech, C-103-100), 10 μg/ml Polymyxin B in water (HiMedia, TC0033), 10 μg/ml Vancomycin in water (MP Biochemicals, 195540), 20 μg/ml Trimethoprim in DMSO (Sigma–Aldrich, 92131), 20 μg/ml Cycloheximide in DMSO (MP Biochemicals, 100183), 20 μg/ml Amphotericin B in DMSO (MP Biomedicals, 195043)]. The bacillary burden was estimated for all the strains for specific time points post-infection by the CFU enumeration method.

**Histopathology**. For histopathology based experiments, the left lung of Mtb infected mice were immersed in 10% Formalin (Sigma–Aldrich, HT401128) for 24 h for fixation and 5 μm thick paraffin-embedded sections were subjected to H and E staining. Tulane National Primate Research Center, USA provided paraffinized blocks of uninfected and Mtb infected macaque lungs. The paraffinized non-Mtb infected and Mtb infected human lungs were acquired from the Postgraduate Institute of Medical Education and Research, Chandigarh, India. All experiments pertaining to human samples were approved by the Institutional Ethics Committee of CSIR-IMTech (Approval no [IEC (Sept 18) #5]). 5 μm thick sections were made and stained with H and E and were visualized under a microscope to understand the pathology. The pathology level was scored by analyzing perivascular cuffing, leukocyte infiltration, multinucleated giant cell formation, and epithelial cell injury using ImageJ software.

**Detection of cellulose by confocal microscopy**. The presence of cellulose was investigated in the samples using staining based methods. Prior to staining, the paraffinized samples were deparaffinized using a serial washing step with xylene and ethanol—1. 100% Xylene for 6 min, 2. Xylene: Ethanol 1:1 for 3 min, 3. 100% Ethanol for 3 min, 4. 95% Ethanol for 3 min, 5. 70% Ethanol for 3 min, 6. 50% Ethanol for 3 min, 7. Distilled water. Samples were stained with CW (3 μM, Sigma–Aldrich), which is specific for β (1,4)-D-glucopyranosyl units present in cellulose. The samples were further stained with CBD-mCh. The samples were also treated with 5 mg/ml Cellulysin Cellulase dissolved in 0.05 M citrate buffer (pH 4.0) for 6 h at 37 °C and then stained with the dyes to check for the presence of cellulose. The presence or absence of bacteria in infected and uninfected samples, respectively, were checked through staining with Phenolic Auramine O-Rhodamine B dye[44]. Samples stained with CBD-mCh were counterstained with DAPI (5 μg/ml) for 5 min to stain the nuclei of the cells. Following staining, the samples were subjected to CLSM using lasers 405 nm for DAPI and Calcofluor White, 488 nm for Auramine, and 561 nm for CBD-mCh.

**Detection of cellulose using raman microscopy**. Raman spectra of 5 μm thick lung sections were acquired using Raman microscope (Renishaw) at CSIR-CSIO, Chandigarh, India. The Raman system was coupled to a microscope with a spatial resolution of nearly 1 micron and a charge-couple device camera for detection. Raman images and spectra were collected after excitation with a 532 nm laser by moving the sample with a motorized stage through a wavenumber span of 800 to 1600 cm$^{-1}$. Commercially available anhydrous cellulose (Sigma) was used as a control. Earlier described Raman spectral profile matched the peaks observed with commercial cellulose[30,58].

**Glucose release upon cellulase treatment**. Glucose release from the lung tissues of mice, macaque, and humans upon treatment with cellulase was estimated using the DNS method of detection of reducing sugars as described earlier[59]. Briefly, the tissue samples were treated with Cellulysin Cellulase (0.5 mg/ml in 0.05 M citrate buffer, pH = 4) for 12 h and to 1 ml of each of the sample, 1 ml of acetate buffer (pH = 4.8) was added along with 3 ml of DNS, and the solution was incubated in a boiling water bath for 5 min. The absorbance was recorded in a spectrophotometer at 540 nm wavelength. Citrate buffer alone was taken as a control.

**Drug susceptibility test in mice**. Eight to ten weeks old C57BL/6 J mice were aerogenically infected with passaged Mtb H37Rv to achieve deposition of around 100 bacilli as described earlier[57]. The infected mice were checked for the enumeration of the bacterial load one month post-infection, following which they were subjected to drug treatment. The animals were administered with a single oral dose by gavage of Rif and INH formulations at concentrations of 80 mg/kg and 70 mg/kg, respectively, for five days a week for four weeks. A group of infected animals was also nebulized with 20U of Cellulysin Cellulase from *Trichoderma viridae* (Merck, 219466), along with Rif and INH treatment. The animals were estimated for their bacterial load, two and four weeks post-treatment after euthanasia. Heat inactivated Cellulysin Cellulase was used as a negative control.

**Confirmation of Mtb in human lungs using fluorescence in situ hybridization (FISH).** FISH to detect Mtb in infected human lungs was performed following published protocols[32,60,61]. Briefly, the paraffinized human lung tissue sections were initially deparaffinized using a serial washing step with xylene and ethanol as previously mentioned, following which the sections were treated with 1 mg/ml Proteinase K and 10 mg/ml Lysozyme in 10 mM Tris (pH 7.5) at 37 °C for 30 min. Next, the samples were incubated in the Prehybridization buffer at 37 °C for 1 h. Prehybridization buffer is composed of 20% 2× SSC, 20% Dextran sulfate, 30% Formamide, 1% 50X Denhardt's reagent, 2.5% of 10 mg/ml PolyA, 2.5% of 10 mg/ml salmon sperm DNA, 2.5% of 10 mg/ml tRNA. The slides were thoroughly washed with 2X SSC buffer. The sections were then incubated in hybridization buffer at 95 °C for 10 min and then chilled on ice for 10 min. Further hybridization was allowed at 37 °C overnight. Hybridization buffer is composed of prehybridization buffer plus 16 SMtbRv probe (5′ FITC – CCACACCGCTAAAG – 3′), which is specific for the 16S rRNA of Mtb at a final concentration of 1 ng/μl. The lung tissue sections were next subjected to a series of washing steps with 1X SSC at room temperature for 1 min, 1× SSC at 55 °C for 15 min, 1× SSC at 55 °C for 15 min, 0.5X SSC at 55 °C for 15 min, 0.5X SSC at 55 °C for 15 min, 0.5× SSC at room temperature for 10 min. The samples were mounted on glass slides and visualized using CLSM with a 488 nm laser.

**Microplate-based alamar blue assay (MABA).** The engineered Mtb strains WT-VC, Rv0062 and Rv1090 were grown in Middlebrook 7H9 media supplemented with 10% ADS (Albumin-Dextrose-Sodium chloride). Exponential cultures were freshly diluted to OD$_{600}$ 0.02 and 50 μl of dilutes cultures were added to 50 μl of media containing 2× MIC concentration of antibiotics (Rif and INH). The assay included appropriate negative (culture without antibiotics) and positive (Rif in case of INH plate and INH in case of Rif plate) controls. The plates were incubated at 37 °C for 7 days. Following incubation, 0.02 ml of 0.02% Resazurin (sodium salt, MP Biomedicals), prepared in sterile 7H9 media was added to each of the wells of the plates and color change was monitored after incubation of approximately 20 h at 37 °C. Excitation was at 530 nm and fluorescence emission was recorded at 590 nm. The efficiency of inhibition by the drugs for each of the three strains was calculated using earlier established methods[62].

**Drug susceptibility testing by CFU.** To examine the sensitivity of the engineered strains of Mtb towards Rif and INH, WT-VC, Rv0062 and Rv1090 were grown to exponential phase (OD$_{600}$ ~0.6) and then diluted to OD$_{600}$ 0.02. The cells were then treated with 1× MIC concentration of Rif and INH. 1 day and 3 days post treatment, the cells were washed with 1× PBS and CFU was estimated using the serial dilution method.

**Estimation of ATP levels.** The ATP levels of each of the three engineered strains were estimated using ATP Bioluminescence Assay kit CLS II (Roche, 11699695001) following the manufacturer's instructions. Briefly, 1 ml of the exponential cultures of each of WT-VC, Rv0062 and Rv1090 were lysed in lysing matrix B tube by bead beating in Tris-EDTA buffer (10 mM Tris, 1 mM EDTA, pH—8.0). 50 μl of the sample supernatant was added to 50 μl of luciferase enzyme, and the chemiluminescence readings were immediately recorded. Cells with 3 h treatment of 1X MIC concentration of Bedaquiline were taken as control to confirm the validity of the assay.

**Estimation of NADH levels.** The NADH levels of each of the three engineered strains were estimated using NAD/NADH quantification kit (Sigma–Aldrich, MAK037) following the manufacturer's instructions. Briefly, 1 ml of the exponential cultures of each of WT-VC, Rv0062 and Rv1090 were lysed in lysing matrix B tube by bead beating in NADH/NAD extraction buffer. 50 μl of the sample supernatants were incubated with 100 μl of Master reaction mix in 96 well plate for 5 mins, following which 10 μl of NADH developer was added in each of the wells and incubated for 3 h. The absorbance readings were taken at 450 nm. The concentrations of NADH for each strain were calculated from the standard curve.

**Availability of materials.** Plasmids encoding CBD-mCh, expression of Rv0062, and Rv1090 will be freely provided after the signing of an MTA.

**Reporting summary.** Further information on research design is available in the Nature Research Reporting Summary linked to this article.

## Data availability
The authors declare that the data supporting the findings of this study are available within the main text of the manuscript and in its supplementary information files. Source data are provided with this paper.

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

## Acknowledgements

We are thankful to Parminder Singh Mavi for the CBD-mCh probe. P.C. and S.B. are funded by SRF from DBT and CSIR, Govt of India, respectively. A.K. is supported by the Swarnajayanti Fellowship (DST/SJF/LSA-02/2016-17) and the National Bioscience Award (BT/HRD-NBA-NWB/37/01/2018) from DST and DBT, Govt of India, respectively. This work was supported through a DST grant to A.K. (DST/SJF/LSA-02/2016-17).

## Author contributions

P.C. and A.K. conceptualized the work. The experiments were conducted by P.C. and S.B., D.K. helped in the acquisition and characterization of non-human primate lung samples. B.D.R. helped in acquiring and characterizing human lung samples. P.C., S.B. and A.K. analyzed data and wrote the manuscript. D.K. and B.D.R. provided critical inputs and improved the manuscript.

## Competing interests

The authors declare no competing interests.
