## [Peer Review File · Nature Communications]

REVIEWER COMMENTS

Reviewer #1 (Remarks to the Author):

The major claims of the paper are that Mycobacteria including NTMs and *M. tuberculosis* form in vitro and in vivo biofilms and that biofilm formation accounts for antimicrobial drug tolerance and immune evasion. Moreover, the investigators demonstrate convincingly that bacteria derived cellulose contributes to the extracellular polymeric substance (EPS) and that enzymatic digestion or overexpression of endogenous cellulase restores antimicrobial drug susceptibility and increases immune susceptibility. The most significant finding is that the development of in vivo biofilms can be demonstrated in animal models (mice and non-human primates) as well as in human tissue sections obtained from patients with TB.

In the reviewer's opinion, this is among the most important and significant studies I have had the opportunity to review and will be of interest to a wide audience and is likely to have a significant impact on the understanding of TB pathogenesis, drug tolerance and immune evasion.

The hypothesis that *Mtb* in particular forms drug tolerant biofilms is a relatively new concept that has not gained wide acceptance across the TB field. The main reason is that until now, nobody has convincingly demonstrated biofilm formation in vivo only in vitro including the reviewer's own studies. These studies are the first to convincingly demonstrate in vivo biofilm formation not only in animal models but also in human post-mortem tissues. Moreover, the investigators demonstrate that bacterial derived cellulose is a major component of the extra cellular matrix that contributes to *Mtb* and NTM biofilm formation and demonstrate the potential for targeting this complex carbohydrate in the development of new diagnostics and therapies.

The studies are thorough, and of high quality and thus convincing. With that being said, the main weakness is the attempt to demonstrate the reversal of immune evasion which lacks specific components of the immune response responsible for clearing infection but it could be argued that this is beyond the scope of this study. However, the strategies used in this study will be critical for future studies that are aimed at specifically dissecting the protective immune response to *Mtb* which is still poorly understood.

There are a couple of important omissions that the reviewer feels would significantly strengthen the manuscript. First of all, these are not the first studies to show that in vitro biofilm formation contributes to the expression of antimicrobial drug tolerance. The authors should cite the studies by A. Ojha et al beginning in 2008 that was among the first group to suggest that *Mtb* biofilm formation contributes to the expression of in vitro drug tolerance. These investigators demonstrated that during pellicle formation, *Mtb* derived free mycolic acids contribute to the EPS of in vitro. Subsequent papers by this group have expanded on these findings and should also be reviewed by the authors.

Secondly, the authors demonstrated that in vitro and in vivo administration of cellulase not only disrupts *Mtb* and NTM biofilm formation but can be administered in vivo to restore drug susceptibility. These are important findings but again are not the first to suggest this therapeutic approach. Previous studies have shown that bacterial and host derived DNA contribute to *Mtb* biofilm formation in vitro, which the authors briefly mention. The excretion of extracellular DNA and RNA has been described to be part of the EPS of a large number of biofilm forming bacteria but less convincingly in mycobacteria. The significance of this is that DNase has been used in vitro and in vivo to target primarily host derived extracellular DNA in the airways of patients with cystic fibrosis. Pulmzyme is FDA approved and is used as an adjunct to potentiate the activity of antimicrobial drugs in CF patients. The importance of this is that the authors fail to mention this important adjunctive therapy and downplay the potential importance of cellulase used as a similar adjunctive treatment which they have demonstrated in these studies.

All in all the manuscript is well written with some minor grammatical errors that need to be corrected.

Reviewer #2 (Remarks to the Author):

General Comments

1. A report of an ambitious and potentially game-changing project to follow up on prior work examining the composition of *M. tuberculosis* biofilms.
2. The sheer scope of the effort and conclusions is a major challenge and apparently the scale of the challenge is not appreciated. The manuscript must convince readers. However, conclusions are made, but lack thorough, critical discussion of alternative explanations, such that the reader is left with the idea that more work is needed.
3. As there is no evidence of genes for polysaccharide and cellulose synthesis in *M. tuberculosis*, this manuscript will face a great deal of scrutiny if published. Thus, it is important to critically review the assumptions of the experimental approaches and the validity of controls. In some instances, experiments are reported without critical examination of substrate specificity of enzymes, fluorescent dyes, and possible other compounds providing signals (e.g., Raman) that are from compounds other than cellulose, but can provide like signals.
4. Please reconsider some assumptions; namely that enzymes only act on specific substrates as polysaccharidases like cellulase are general esterases and can hydrolyze bonds in a wide variety of polysaccharides.

Specific Comments

Title. Inappropriate as it miss-characterizes the data in the manuscript. Please consider a title, "Role of cellulose in *M. tuberculosis* biofilms in lungs."

Introduction

Line 4. Please add a reference for treatment failures even with higher doses of antibiotics"

Line 5. Do you mean to include "leprosy"?

Line 13. Were the biofilms formed in animals, tissue, or inanimate surfaces?

Lines 1-2. Please provide a reference to overexpression of cellulase preventing pellicle formation.

Lines 2-13. This section argues against the object of the manuscript to implicate the role of cellulose in *M. tb* biofilms. Yet, sadly, there is no discussion of this problem, save "possibility of a non-canonical cellulose synthesis system in *Mtb* cells." As this is a key point, I am reminded of the controversy generated by a recent paper on spores in mycobacteria in the absence of any spore-associated genes. As a novel and challenging topic is raised by the manuscript, please consider that every test and result will be closely scrutinized. As written, the manuscript lacks that critical examination of the experimental approaches and results.

Lines 7-10. The sentence is compound and needs to be separated into, perhaps, three sentences.

Results

Page 5, Line 3 – Page 14, line 12. There does not seem to be a clear pattern to the inclusion of data as Figures in the text or as supplements. In some cases, please consider placing some supplementary figures in the text. For example, this is needed when the objective is to prove an important point (e.g., Fig 5 and others).

Page 5, Line 13. Delete "Alternatively" as it is inappropriate.

Page 5, line 14. Please describe the "cultures".

Page 5, line 17. Delete "interestingly"

Page 5, lines 20-23. This information stands in contrast to all other published data concerning biofilm formation; all major Mycobacterium species readily form biofilms in the absence of DTT or any reducing agent. What is unique in the experimental system that was used, that led to the requirement for DTT? Are the surfaces modified?

Page 6, lines 1-5. What is missing is consideration of the effect of the "thiol reductive stress on the antibiotic and its action, not just on a biofilm.

Page 6, lines 6-8. The literature presents quite a bit about mycobacterial biofilm composition. I note a paper from Graham Hatful's lab showing the presence of lipid in mycobacterial biofilms is not included.

Page 6, lines 13-17. Please consider the substrate specificity of Texas Red as arabinomannan is a major constituent of the M. avium outer membrane. The absence of any consideration of cross-reactions or the lack of specificity of the tools used leads the reader to consider other explanations.

Page 6, lines 17-23. Why is the information in a Supplementary Figure and not in the text?

Page 7, line 1. Was DTT present along with the enzymes? Was any consideration given to the effect of DTT on the enzymes and their activity?

Page 7, line 16. Why is this figure in the supplementary information, as it offers standard methods for visualization of mycobacteria in tissue as a comparison for the approaches used in the manuscript.

Page 7, line 20. Delete "Interestingly"

Page 7, line 20 – Page 9, line 15. This section appears to present a re-discovery of the "electronic transparent zone" (ETZ) that was described for M. tuberculosis, most notably in papers by Nalin Rastogi and Hugo David at the Institut Pasteur. It is quite possible that the authors of this work have also discovered the ETZ, now with a different role or providing details of its composition. Needless, the information in this section needs to consider the ETZ.

Page 11, line 4 – Page 12, line 7. There is evidence in this section that well-recognized methods for Mtb visualization are used. However, there is no side-by-side comparison between those results and those involving cellulose and alternative staining approaches. As such the data do not convince.

Page 12, line 8 – Page 13, line 9. Here is an example of the ambitious nature of the work and its description. There is no critical evaluation of the experimental approaches and assumptions of the methods and their results. For example, lines 19-21 on page 12; "these strains were attenuated in biofilm formation in response to DTT (Fig. 5f-g). There is simply neither numerical data suitable for correlation statistical analysis, nor a critical analysis of the methodology and alternative possible explanations.

Page 13, line 10 – page 14, line 14. Again, another example of the broad scope of the presentation and here, for example, there is no consideration of the effect of the treatments on the antibiotics and the metabolism of the cells; a narrow perspective is taken.

Discussion

Page 15, line 2 – Page 17, line 19. There is no critical analysis of the methods or of alternative explanations. In their absences, the reader is not convinced with the conclusions made by the authors.

Reviewer #3 (Remarks to the Author):

In their paper entitled "Mycobacterium tuberculosis biofilms in lungs protect resident bacilli from the host immune system and antimycobacterial agents", Chakraborty et al. first show that cellulose is an important component of mycobacterial (including Mtb) biofilms, and then use this as a biomarker, as well as other techniques, to study the presence of Mtb biofilms in the lungs of

Mtb-infected mice, nonhuman primates and humans. They also reported that Mtb strains defective for generating biofilms show reduced growth in the lungs of mice during the acute phase of infection and treatment with nebulized cellulase increases the antimycobacterial activity of isoniazid and rifampicin. From these data, they conclude that biofilms play an important role in Mtb escape from killing by host immunity and by standard anti-TB treatment. This is certainly a provocative paper, as the significance of biofilms in Mtb persistence/antibiotic tolerance in vivo is controversial and highly debated in the field. In general, the studies are thorough, with multiple complementary methodologies used to support the presence of biofilms in Mtb-infected lung tissues from three different species (mice, macaques and humans). However, there are several noted weaknesses, as outlined below.

Specific comments:

1. Figure 1: Abbreviations should be defined in the figure legends.
2. Supplementary Figure 5 (page 7, lines 1-2): It is stated that "We did not observe biofilm formation in the presence of Cellulase and Proteinase K", however there is still some crystal violet staining after these treatments. It is important to have a planktonic culture control to compare CV values with Cellulase- and Proteinase K-treated biofilms.
3. Figure 2 and Supplementary Figure 8: It is stated that Calcofluor White (CW) stains polysaccharides containing $\beta(1\rightarrow4)$ and $\beta(1\rightarrow3)$ glycosidic bonds with high specificity. For this reason, it was used to determine the presence of biofilms in granulomas. Why was there CW staining in the lung tissues from uninfected mice? Also, after treatment with cellulase, there is still quite high CW staining signal (much higher than uninfected background control). Similarly, cellulose staining with CBD-mCh was reduced after treatment of the tissue block with cellulase, but not completely. Is this because of incomplete enzymatic activity in degrading biofilms or because of nonspecific CW and/or CBD-mCh binding?
4. Figures 2 and 3: Microscopy fields lacking Mtb cells (by acid-fast bacilli (AFB) staining) did not show any CW staining in mouse lungs. Similarly, microscopy fields lacking Mtb cells in the Mtb-infected macaque lung sections did not stain for cellulose. We know that tissues stain positive for AFB when there are $> 10^4$ bacilli/gram of tissue. Does this mean that smaller populations of bacilli (that are widely believed to be responsible for latent TB infection) do not form biofilms?
5. In the NHP model and in the human Mtb-infected tissues (Figures 3 and 4), the CW signal is not as high as in this mouse model (Figure 2). Why is this? Also, how are the CW and CBDmCh signals normalized (e.g., per gram tissue)? In Figures 2-4, the "cellulase treated" group should be labeled "Infected/cellulase treated". For the CW and CBDmCh panels, the X-axis could be improved by labeling as follows:
Mtb infection - + +
Cellulase treatment - - +
6. Figure 4g: If cellulose is a specific biomarker for biofilms, why were uninfected human lung tissues releasing glucose after cellulase treatment?
7. Figure 5: Although attenuation of the two Mtb mutants are shown during in vivo infection but not in vitro, data are not shown for the respective complemented strains.
8. Figure 5 (page 13, lines 8-9): Although the dogma is that adaptive immunity begins 3 weeks post-infection, it is difficult to speculate about the role of innate vs. adaptive immune system based on the data presented. It seems more prudent to simply state that these Mtb genes are required during acute infection.
9. Discussion (pages 15-16): The three populations described are speculative. There is also the possibility that extracellular bacilli may survive in the necrotic core of granulomas, independent of biofilms.
10. Change "mice" to "mouse" throughout when used as an adjective. For example, "mice lungs" and "mice model" (page 8, lines 1-2).

A point-by-point response to the reviewer's concerns/suggestions.

Authors are thankful to the reviewers for incisively reviewing the manuscript, emphasizing the manuscript's strengths, and articulating its weakness. The reviewers have been very kind and have suggested some experiments to eliminate the current shortcomings and to strengthen the data presented vis-à-vis conclusions derived in the manuscript. We have performed several experiments to address the issue raised by the reviewers including, DTNB assay to determine the intracellular thiols levels in *Mycobacterium avium*, *Mycobacterium abscessus*, and *Mycobacterium fortuitum* upon exposure to 6 mM DTT, analyzed the enzyme activities post-exposure of *M. avium*, *M. abscessus*, and *M. fortuitum* cells to DTT to study the effects of residual DTT (if any) on enzymes, analyzed newly engineered strains for their metabolic rate and sensitivity towards the antimycobacterials, repeated a few experiments with the suggested controls. Reviewers have also suggested a few changes in the manuscript text and rearranging the manuscript's data. We believe that incorporating reviewers' suggestions and new data has significantly improved the manuscript and hope that it has evolved to become acceptable for publication. The suggestions and questions raised by the reviewer/s are addressed below.

Reviewer #1 (Remarks to the Author):

The major claims of the paper are that Mycobacteria including NTMs and *M. tuberculosis* form in vitro and in vivo biofilms and that biofilm formation accounts for antimicrobial drug tolerance and immune evasion. Moreover, the investigators demonstrate convincingly that bacteria derived cellulose contributes to the extracellular polymeric substance (EPS) and that enzymatic digestion or overexpression of endogenous cellulase restores antimicrobial drug susceptibility and increases immune susceptibility. The most significant finding is that the development of in vivo biofilms can be demonstrated in animal models (mice and non-human primates) as well as in human tissue sections obtained from patients with TB.

In the reviewers opinion, this is among the most important and significant studies I have had the opportunity to review and will be of interest to a wide audience and is likely to have a significant impact on the understanding of TB pathogenesis, drug tolerance and immune evasion.

Response: We are thankful to the reviewer for agreeing that the findings reported here are of considerable interest to the field.

The hypothesis that *Mtb* in particular forms drug-tolerant biofilms is a relatively new concept that has not gained wide acceptance across the TB field. The main reason is that until now, nobody has convincingly demonstrated biofilm formation in vivo only in vitro including the reviewers own studies. These studies are the first to convincingly demonstrate in vivo biofilm formation not only in animal models but also in human post-mortem tissues. Moreover, the investigators demonstrate that bacterial derived cellulose is a major component of the extra cellular matrix that contributes to *Mtb* and NTM biofilm formation and demonstrate the potential for targeting this complex carbohydrate in the development of new diagnostics and therapies.

Response: We are thankful to the reviewer for appreciation of the study. These comments are

encouraging and reassuring that we have been moving in the right direction.

The studies are thorough, and of high quality and thus convincing. With that being said, the main weakness is the attempt to demonstrate the reversal of immune evasion which lacks specific components of the immune response responsible for clearing infection but it could be argued that this is beyond the scope of this study. However, the strategies used in this study will be critical for future studies that are aimed at specifically dissecting the protective immune response to Mtb which is still poorly understood.

Response: We understand the concern of the reviewer. We agree with the reviewer that this study only provides a glimpse that the biofilms protect from the immune system. However, a detailed analysis of biofilms mediated evasion of innate and adaptive immune response is beyond the scope of this manuscript. We have discussed the same in the discussion section of the revised manuscript. This manuscript is primarily focused on the demonstration of Mtb biofilms *in vivo*. Furthermore, data suggest that these biofilms protect the resident bacilli from the host and the antimycobacterial agents.

There are a couple of important omissions that the reviewer feels would significantly strengthen the manuscript. First of all, these are not the first studies to show that *in vitro* biofilm formation contribute to the expression of antimicrobial drug tolerance. The authors should cite the studies by A. Ojha et al beginning in 2008 that was among the first group to suggest that Mtb biofilm formation contributes to the expression of *in vitro* drug tolerance. These investigators demonstrated that during pellicle formation, Mtb derived free mycolic acids contribute to the EPS of *in vitro*. Subsequent papers by this group have expanded on these findings and should also be reviewed by the authors.

Response: We are thankful to the reviewer for pointing out this issue. We have incorporated these references in the revised manuscript. We have also clearly suggested the reference (Mol Microbiol. 2008 Jul; 69(1): 164–174) was amongst the first studies to demonstrate the role of *in vitro* biofilms in drug tolerance. We have also reviewed other papers by the group in the introduction of the revised manuscript.

Secondly, the authors demonstrated that *in vitro* and *in vivo* administration of cellulase not only disrupts Mtb and NTM biofilm formation but can be administered *in vivo* to restore drug susceptibility. These are important findings but again are not the first to suggest this therapeutic approach. Previous studies have shown that bacterial and host derived DNA contribute to Mtb biofilm formation *in vitro*, which the authors briefly mention. The excretion of extracellular DNA and RNA has been described to be part of the EPS of a large number of biofilm forming bacteria but less convincingly in mycobacteria. The significance of this is that DNase has been used *in vitro* and *in vivo* to target primarily host derived extracellular DNA in the airways of patients with cystic fibrosis. Pulmzyme is FDA approved and is used as an adjunct to potentiate the activity of antimicrobial drugs in CF patients. The importance of this is that the authors fail to mention this important adjunctive therapy and downplay the potential importance of cellulase used as a similar adjunctive treatment which they have demonstrated in these studies.

Response: The reviewer has correctly pointed out that we have downplayed the significance of the use of cellulase as an adjunct therapy. We plan to emphasize our findings that cellulase administration could reduce bacterial load by the frontline antimycobacterial agents in the discussion section of the revised manuscript. We have emphasized the extracellular nucleic acids are an important component of EPS of mycobacterial biofilms.

All in all the manuscript is well written with some minor grammatical errors that need to be corrected.

Response: We are thankful to the reviewer for encouraging comments. We have corrected the grammatical errors in the revised manuscript.

Reviewer #2 (Remarks to the Author):

General Comments

1. A report of an ambitious and potentially game-changing project to follow up on prior work examining the composition of *M. tuberculosis* biofilms.
2. The sheer scope of the effort and conclusions is a major challenge and apparently the scale of the challenge is not appreciated. The manuscript must convince readers. However, conclusions are made, but lack thorough, critical discussion of alternative explanations, such that the reader is left with the idea that more work is needed.
3. As there is no evidence of genes for polysaccharide and cellulose synthesis in *M. tuberculosis*, this manuscript will face a great deal of scrutiny if published. Thus, it is important to critically review the assumptions of the experimental approaches and the validity of controls. In some instances, experiments are reported without critical examination of substrate specificity of enzymes, fluorescent dyes, and possible other compounds providing signals (e.g., Raman) that are from compounds other than cellulose, but can provide like signals.
4. Please reconsider some assumptions; namely that enzymes only act on specific substrates as polysaccharidases like cellulose are general esterases and can hydrolyze bonds in a wide variety of polysaccharides.

Response: We are thankful to the reviewer for appreciating the importance of the scope of the manuscript. The reviewer has also raised concerns on the details provided about the methodology used, data interpretation, conclusions drawn, and the absence of alternative explanations. In the light of these comments, we have provided more details on methods used, justification of conclusions drawn, and have presented alternative hypotheses wherever required in the revised manuscript.

Specific Comments

Title. Inappropriate as it miss-characterizes the data in the manuscript. Please consider a title, "Role of cellulose in *M. tuberculosis* biofilms in lungs."

Response – We respectfully differ with the reviewer on this. There are two main findings of the study

- (i) *M. tuberculosis* forms biofilms during pulmonary infection
- (ii) Biofilms protect bacteria from host defense and antimycobacterial agents

We believe that the most appropriate title will be "*Mycobacterium tuberculosis* biofilms in lungs; protection from host defense and antimycobacterials." Thus, we have changed the title in the revised manuscript.

I would like to emphasize that we have earlier demonstrated cellulose's role in TB biofilms (Thiol reductive stress induces cellulose-anchored biofilm formation in *Mycobacterium tuberculosis*. *Nat Commun.* 2016 Apr 25;7:11392. doi: 10.1038/ncomms11392). Thus it will be most appropriate that the title emphasizes the role of biofilms in protecting the bacterial cells from host and

antibiotics instead of the role of cellulose in biofilms. We hope that the reviewer understands our point of view on this.

Introduction

Line 4. Please add a reference for treatment failures even with higher doses of antibiotics”

Response – We thank the reviewer for pointing out the error. The suitable reference (APMIS. 2017 Apr;125(4):304-319. doi: 10.1111/apm.12673) is added in the revised manuscript.

Line 5. Do you mean to include “leprosy”?

Response – We thank the reviewer for the query. Leprosy, caused by *M. leprae*, shows certain features of biofilm infections, such as slow response to drugs, the tendency for recurrence of disease after treatment, and the presence of amyloid-like structures at the infection sites (JAMA Dermatol. 2017 Mar 1;153(3):261-262. doi: 10.1001/jamadermatol.2016.5506).

Line 13. Were the biofilms formed in animals, tissue, or inanimate surfaces?

Response – We understand the concern of the reviewer. In this study, the authors studied *M. tuberculosis* biofilms formed on the tissue culture plates in the presence of leukocyte lysate. We have now revised the manuscript to reflect this.

Lines 1-2. Please provide a reference to overexpression of cellulose preventing pellicle formation.

Response – We thank the reviewer for pointing out this. The appropriate reference (Glycobiology. 2017 May 1;27(5):392-399. doi: 10.1093/glycob/cwx014) is now added to the revised manuscript.

Lines 2-13. This section argues against the object of the manuscript to implicate the role of cellulose in *M. tb* biofilms. Yet, sadly, there is no discussion of this problem, save “possibility of a non-canonical cellulose synthesis system in *Mtb* cells.” As this is a key point, I am reminded of the controversy generated by a recent paper on spores in mycobacteria in the absence of any spore-associated genes. As a novel and challenging topic is raised by the manuscript, please consider that every test and result will be closely scrutinized. As written, the manuscript lacks that critical examination of the experimental approaches and results.

Response – We understand the concerns of the reviewer. In the revised manuscript, we will describe the methods utilized in the earlier study to conclude that cellulose is a component of *M. tuberculosis* biofilms. We agree that given the novel and challenging topic raised by the manuscript, every result described here will be scrutinized by the community at large. In light of this, several approaches were our previous study and in this study to demonstrate the presence of cellulose in *Mtb* biofilms, and now cellulose encased *Mtb* biofilms in the lungs. We are confident about these findings. Fortunately, a study following our study demonstrating the presence of cellulose in the *M. tuberculosis* biofilms Laurent Kremer and co-workers treated mycobacterial pellicle biofilms with mycobacterial cellulase. They detected the release of cellobiose using ionic chromatography. Furthermore, in line with the strains generated in the current study, the group demonstrated that overexpression of cellulase in *M. smegmatis* results in the attenuation of the mycobacterial capability to form pellicle biofilms. However, considering

the reviewer's concerns, we have critically discussed experimental approaches and provided alternative explanations.

Lines 7-10. The sentence is compound and needs to be separated into, perhaps, three sentences.

Response – We appreciate the reviewer's concern. The sentence has been reframed as - Several mycobacterial species, such as *M. neoaurum*, *M. cosmeticum*, etc., contain genes encoding components of the cellulose synthase. This raises the possibility mycobacterial species could use cellulose as a structural component of biofilms. However, the Mtb genome does not seem to encode either homolog of canonical BcsA or BcsB.

Results

Page 5, Line 3 – Page 14, line 12. There does not seem to be a clear pattern to the inclusion of data as Figures in the text or as supplements. In some cases, please consider placing some supplementary figures in the text. For example, this is needed when the objective is to prove an important point (e.g., Fig 5 and others).

Response – We understand the reviewer's concerns, and in the revised manuscript, we have rearranged the data. This has resulted in moving a lot of important data in the main figures from the supplementary data. We believe that this has improved the presentation of the data and the readability of the manuscript.

Page 5, Line 13. Delete “Alternatively” as it is inappropriate.

Response – We thank the reviewer for the suggestion. The word “Alternatively” has been deleted.

Page 5, line 14. Please describe the “cultures”.

Response – We are thankful to the reviewer for the query. The “cultures,” as pointed out by the reviewer, refers to the exponential cultures of Mav, Mab and Mfo, which had been exposed to 6 mM DTT followed by mild shaking. These details are now provided in the revised manuscript.

Page 5, line 17. Delete “interestingly”

Response – We thank the reviewer for the suggestion. The word “interestingly” has been changed to “however” in the revised manuscript.

Page 5, lines 20-23. This information stands in contrast to all other published data concerning biofilm formation; all major Mycobacterium species readily form biofilms in the absence of DTT or any reducing agent. What is unique in the experimental system that was used, that led to the requirement for DTT? Are the surfaces modified?

Response – We understand the concern of the reviewer. We agree that major mycobacterial species readily form biofilms and could adhere to a range of substratum. However, the effect of thiol reductive stress on biofilms in many species is not studied. We have earlier demonstrated that thiol-reductive stress induced by DTT results in surface adherent Mtb biofilms. The effect of thiol reductive stress in other species and its relationship with biofilm formation is not well characterized. Thus here, we analyzed whether thiol reductive stress induced by DTT led to biofilms formation or not. The assumption was that such a phenotype might be limited to *M. tuberculosis*. But we observed that thiol reductive stress-induced biofilms formation in all the

three species detected. Notably, just incubation of these species with thiol reductant BME or with oxidized DTT did not induce biofilm formation. In line with this, we observed that only DTT causes intracellular thiol reductive stress. Thus the biofilm formation is closely linked with the reductive stress in Mac, Mab, and Mfo. The genetic regulation of this pathway and molecular events associated with this phenotype is yet to be characterized.

We would also like to bring it to the reviewer's notice that DTT does not modify the surfaces. As a control, we have used beta-mercaptoethanol. This thiol reductant does not enter inside that mycobacterial cell and thus is unable to induce biofilm formation. On the other hand, DTT leads to intracellular thiol reductive stress and promotes biofilm formation. To further answer these questions, we now plan to measure the intracellular redox levels using the DTNB assay. Data is now added to the manuscript. We have now extensively modified the results section describing the biofilms of Mav, Mab, and Mfo.

Page 6, lines 1-5. What is missing is consideration of the effect of the “thiol reductive stress on the antibiotic and its action, not just on a biofilm.

Response – We understand the concern of the reviewer. Quite a few studies have analyzed the role of thiols in survival against the host generated stress. The Group of Robert Abramovitch at Michigan State University has demonstrated that depletion of cellular thiol pool using compound AC2P36 is associated with enhanced mycobacterial killing by antimycobacterials (Cell Chem Biol. 2017 Aug 17; 24(8): 993–1004.e4). On the contrary, the effect of the accumulation of reduced intracellular thiol on antibiotics remains undeciphered. However, this manuscript aims to determine Mtb biofilms' formation during infection besides analyzing biofilms' role in protection from the host defenses and antimycobacterials. Thus, we feel that examining the role of thiol reductive stress on antibiotics and its action is beyond this manuscript's scope.

Page 6, lines 6-8. The literature presents quite a bit about mycobacterial biofilm composition. I note a paper from Graham Hatful's lab showing the presence of lipid in mycobacterial biofilms is not included.

Response – We are thankful to the reviewer for pointing out this inadvertent error. We have now included the appropriate references in the manuscript and modified the text in the manuscript.

Page 6, lines 13-17. Please consider the substrate specificity of Texas Red as arabinomannan is a major constituent of the M. avium outer membrane. The absence of any consideration of cross-reactions or the lack of specificity of the tools used leads the reader to consider other explanations.

Response – We understand the reviewer's concern and agree that Texas Red could bind to polysaccharides like arabinomannan. However, in the confocal images, we have observed extensive Texas red staining in extracellular spaces. These spaces are far away from the mycobacterial cells. Thus, these data suggest that large quantities of polysaccharides are present between bacterial cells, and they are perhaps the components of the extracellular matrix. We have now included these considerations in the revised manuscript.

Page 6, lines 17-23. Why is the information in a Supplementary Figure and not in the text?

Response – We are thankful to the reviewer for raising this issue. In the revised manuscript, we have moved a lot of information from the main text's supplementary data.

Page 7, line 1. Was DTT present along with the enzymes? Was any consideration given to the effect of DTT on the enzymes and their activity?

Response – We are thankful to the reviewer for raising this concern. We would like to clarify that enzymes were added to the biofilms after washing them thoroughly with PBS. Thus, DTT was not present along with the enzymes. Furthermore, we have analyzed enzyme activity at the end of the assay to ensure that the enzymes were active during the assay.

Page 7, line 16. Why is this figure in the supplementary information, as it offers standard methods for visualization of mycobacteria in tissue as a comparison for the approaches used in the manuscript.

Response – We are thankful to the reviewer for pointing this out. This figure is now presented as the main figure.

Page 7, line 20. Delete “Interestingly”

Response – We thank the reviewer for the suggestion. The word “interestingly” is deleted in the revised manuscript.

Page 7, line 20 – Page 9, line 15. This section appears to present a re-discovery of the “electronic transparent zone” (ETZ) that was described for *M. tuberculosis*, most notably in papers by Nalin Rastogi and Hugo David at the Institut Pasteur. It is quite possible that the authors of this work have also discovered the ETZ, now with a different role or providing details of its composition. Needless, the information in this section needs to consider the ETZ.

Response – We understand the concern of the reviewer. There is a possibility that the layer of polysaccharide, which we believe also contains cellulose, could form an “electron transparent zone” (ETZ) around the mycobacterial cells in the tissue. We have discussed the same in the discussion now and mentioned it in the revised manuscript's result section.

Page 11, line 4 – Page 12, line 7. There is evidence in this section that well-recognized methods for *Mtb* visualization are used. However, there is no side-by-side comparison between those results and those involving cellulose and alternative staining approaches. As such the data do not convince.

Response – We understand the concern of the reviewer. We have utilized Phenolic Auramine O – Rhodamine B dye for the visualization of *Mtb* cells. The cellulose was visualized using a CBD-mCherry probe as well as calcofluor white staining. Only finding cellulose around the mycobacterial infection site is the new information, which was confirmed using alternative approaches. Thus our purpose was not to quantitate and compare the staining of *Mtb* cells vs. the cellulose. We believe that evidence emerging from several methods strongly indicates the presence of cellulose around bacterial cells.

Page 12, line 8 – Page 13, line 9. Here is an example of the ambitious nature of the work and its description. There is no critical evaluation of the experimental approaches and assumptions of the methods and their results. For example, lines 19-21 on page 12; “these strains were attenuated in biofilm formation in response to DTT (Fig. 5f-g). There is simply neither numerical data suitable for correlation statistical analysis, nor a critical analysis of the methodology and alternative possible explanations.

Response – We understand the concern of the reviewer. This section was written to provide evidence suggesting that cellulase overexpressing strains are attenuated for biofilm formation in response to intracellular reductive stress induced by DTT. Figure 5F was used to demonstrate that Mtb overexpressing cellulase Rv0062 or Rv1090 does not form biofilms in response to DTT. Figure 5g (now Figure. 8g) provided details of the quantification of DTT induced biofilms formation in these strains through the use of crystal violet staining. To convince the reviewer, we characterized the strains for their metabolism and drug sensitivity (Supplementary Fig. 12). We have changed the panel of figure 5F (now Figure 8f) and explained these data in more detail. Furthermore, these strains are altered for pellicle formation; their colony morphology differs from the vector control. In the light of these data, it is reasonable to conclude that these strains are deficient in biofilm formation. We would like to emphasize that the methodology used for these experiments was earlier described (Nature Communication 2016 7, Article number: 11392). We have elaborated on the method used in the revised manuscript.

Page 13, line 10 – page 14, line 14. Again, another example of the broad scope of the presentation and here, for example, there is no consideration of the effect of the treatments on the antibiotics and the metabolism of the cells; a narrow perspective is taken.

Response – We respect the point of view of the reviewer; however, we disagree. In these experiments, the only variable is the administration of the nebulized cellulase or heat-inactivated nebulized cellulase to the mice. The rest of the parameters, including antibiotics and mice strain, etc., all were the same. Furthermore, the presence of cellulase in in vitro cultures does not inhibit the growth of the Mtb cells. Thus, we believe it is reasonable to conclude that cellulase treatment, which disintegrates the Mtb biofilms, leads to improvement in bacterial killing by RIF and INH.

Discussion

Page 15, line 2 – Page 17, line 19. There is no critical analysis of the methods or of alternative explanations. In their absences, the reader is not convinced with the conclusions made by the authors.

Response – We have included a critical analysis of the methods and considered alternative explanations in the revised manuscript.

Reviewer #3 (Remarks to the Author):

In their paper entitled “Mycobacterium tuberculosis biofilms in lungs protect resident bacilli from the host immune system and antimycobacterial agents”, Chakraborty et al. first show that cellulose is an important component of mycobacterial (including Mtb) biofilms, and then use this as a biomarker, as well as other techniques, to study the presence of Mtb biofilms in the lungs of Mtb-infected mice, nonhuman primates and humans. They also reported that Mtb strains defective for generating biofilms show reduced growth in the lungs of mice during the acute phase of infection and treatment with nebulized cellulase increases the antimycobacterial activity of isoniazid and Rifampicin. From these data, they conclude that biofilms play an important role in Mtb escape from killing by host immunity and by standard anti-TB treatment. This is certainly a provocative paper, as the significance of biofilms in Mtb persistence/antibiotic tolerance in vivo is

controversial and highly debated in the field. In general, the studies are thorough, with multiple complementary methodologies used to support the presence of biofilms in Mtb-infected lung tissues from three different species (mice, macaques and humans). However, there are several noted weaknesses, as outlined below.

Response: We are thankful to the reviewer for judiciously reviewing the manuscript, emphasizing the manuscript's strengths, and bring out the manuscript's weakness. We are glad to learn that the reviewer is enthusiastic about the findings reported in this manuscript.

Specific comments:

1. Figure 1: Abbreviations should be defined in the figure legends.

Response – We are thankful to the reviewer for this suggestion. The abbreviations for Mav (*Mycobacterium avium*), Mab (*Mycobacterium abscessus*), and Mfo (*Mycobacterium fortuitum*) have been incorporated in the figure legend.

2. Supplementary Figure 5 (page 7, lines 1-2): It is stated that “We did not observe biofilm formation in the presence of Cellulase and Proteinase K”, however there is still some crystal violet staining after these treatments. It is important to have a planktonic culture control to compare CV values with cellulase- and Proteinase K-treated biofilms.

Response – We are thankful to the reviewer for pointing out this issue. In our opinion, the crystal violet staining obtained after the individual enzyme treatment, both 3 hours and 29 hours post DTT exposure, may be arising from the residual biofilm material that remains on the plate. However, as the reviewer suggested, we have now included a planktonic control in experiments.

3. Figure 2 and Supplementary Figure 8: It is stated that Calcofluor White (CW) stains polysaccharides containing $\beta(1\rightarrow4)$ and $\beta(1\rightarrow3)$ glycosidic bonds with high specificity. For this reason, it was used to determine the presence of biofilms in granulomas. Why was there CW staining in the lung tissues from uninfected mice? Also, after treatment with cellulase, there is still quite high CW staining signal (much higher than uninfected background control). Similarly, cellulose staining with CBD-mCh was reduced after treatment of the tissue block with cellulase, but not completely. Is this because of incomplete enzymatic activity in degrading biofilms or because of nonspecific CW and/or CBD-mCh binding?

Response – We understand the reviewer's concern. The stains like Calcofluor White and VBD-mCherry have a very high affinity for polysaccharides containing $\beta(1\rightarrow4)$ and $\beta(1\rightarrow3)$ glycosidic bonds with high specificity. In line with these, confocal images for CW and CBD-mCherry staining and the respective quantitation suggest that very little or no staining in the uninfected mouse lungs. Whatever little signal is detected is due to the background that is always observed in confocal microscopy, and the same is reflected in the quantitation. Also, after cellulase treatment, the signals are higher than the uninfected lungs (Uninfected lungs having ~ 1 AU, Infected lungs having ~ 800 AU, and cellulase treated infected lungs having ~200 AU signal units for CW staining). This, to our speculation, is due to the residual cellulose remaining after the cellulase treatment, which may also arise due to partial cellulose degradation. In fact, enzymes do not work efficiently on the tissue blocks as they work in the solution. The same is the case for CBD-mCherry. We plan to provide an explanation regarding this in the result as well in the discussion section.

4. Figures 2 and 3: Microscopy fields lacking Mtb cells (by acid-fast bacilli (AFB) staining) did not show any CW staining in mouse lungs. Similarly, microscopy fields lacking Mtb cells in the Mtb-infected macaque lung sections did not stain for cellulose. We know that tissues stain positive for

AFB when there are > 10⁴ bacilli/gram of tissue. Does this mean that smaller populations of bacilli (that are widely believed to be responsible for latent TB infection) do not form biofilms?

Response – We thank the reviewer for pointing out this critical issue. The AFB detection threshold in human sputum samples ranges from 10⁴-10⁵ bacilli per ml (J Clin Diagn Res. 2014 Jul;8(7):ZC42-5. doi: 10.7860/JCDR/2014/9764.4587). However, the specificity and sensitivity of Auramine O – Rhodamine B is around 10 % higher than AFB. Accordingly, we found greater staining signals for CW where the bacillary burden was more elevated and less or no staining where there was little or no bacillary burden, as stained by Auramine O – Rhodamine B. In this manuscript, we focussed on investigating the presence of Mtb biofilms during infection in animals and humans and whether such biofilms protect resident bacilli from host defenses and antimycobacterial agents.

To investigate biofilms' relevance, vis-a-vis active and latent TB requires lung tissue samples from the Cornell mice model of dormancy and non-human primates having latent TB. Such studies take time in the magnitude of a couple of years. But these studies are beyond the scope of this study, and we plan to work on this exciting area of research in the future.

5. In the NHP model and in the human Mtb-infected tissues (Figures 3 and 4), the CW signal is not as high as in this mouse model (Figure 2). Why is this? Also, how are the CW and CBDmCh signals normalized (e.g., per gram tissue)? In Figures 2-4, the “cellulase treated” group should be labeled “Infected/cellulase treated”. For the CW and CBDmCh panels, the X-axis could be improved by labeling as follows:

Mtb infection - + +

Cellulase treatment - - +

Response – We are thankful to the reviewer for his insightful comment. We have utilized the mean pixel intensities for normalization against individual ROIs based on fixed areas for each of the samples. These are standard methods of comparison in confocal microscopy. We have changed the labeling of the samples, as suggested by the reviewer.

6. Figure 4g: If cellulose is a specific biomarker for biofilms, why were uninfected human lung tissues releasing glucose after cellulase treatment?

Response – We understand the concern of the reviewer. We would like to bring it to the reviewer's notice that the amount of glucose released from the uninfected human lung after cellulase treatment almost equates to the same released from citrate buffer control in the mentioned figure. Citrate buffer is used in cellulase assays.

7. Figure 5: Although attenuation of the two Mtb mutants are shown during in vivo infection but not in vitro, data are not shown for the respective complemented strains.

Response – Although the reviewer's concern is paramount in principle, we think there has been some misunderstanding in getting our point through. Our manuscript has used two cellulase overexpressing Mtb strains and not Mtb knock out mutant strains, as mentioned by the reviewer.

8. Figure 5 (page 13, lines 8-9): Although the dogma is that adaptive immunity begins 3 weeks post-infection, it is difficult to speculate about the role of innate vs. adaptive immune system based on the data presented. It seems more prudent to simply state that these Mtb genes are required during acute infection.

Response – We are thankful to the reviewer for this suggestion. We have revised the manuscript's results and the discussion sections to reflect on this in the revised manuscript.

9. Discussion (pages 15-16): The three populations described are speculative. There is also the possibility that extracellular bacilli may survive in the necrotic core of granulomas, independent of biofilms.

Response – We agree with the reviewer and changed part of the discussion accordingly.

10. Change “mice” to “mouse” throughout when used as an adjective. For example, “mice lungs and “mice model” (page 8, lines 1-2).

Response – We thank the reviewer for his suggestion. The changes are made in the revised manuscript.

REVIEWERS' COMMENTS

Reviewer #1 (Remarks to the Author):

In the reviewers opinion, the authors have adequately addressed all the reviewers concerns and comments. In response to the reviewers, the additions, omissions and changes to the manuscript have only strengthened the authors conclusions. These data are a valuable contribution to the field and addresses important unanswered questions related to the importance of in vivo and in vitro mycobacterial biofilms in pathogenesis and drug tolerance and resistance.

Randall J. Basaraba

Reviewer #2 (Remarks to the Author):

The manuscript has been appropriately revised following the guidance of the Reviewers

Reviewer #3 (Remarks to the Author):

[No further comments for authors]